# MoCa: Modeling Object Consistency for 3D Camera Control in Video Generation

**Zhijing Cheng**[1,2]* **Xuancheng Zhang**[2] **Donglin Di**[2] **Chen Wei**[2] **Hao Li**[2] **Xun Yang**[1]†

[1]University of Science and Technology of China      [2]Li Auto

`mumucc@mail.ustc.edu.cn`      `xyang21@ustc.edu.cn`

`{zhangxuancheng, didonglin, lihao43, chenwei10}@lixiang.com`

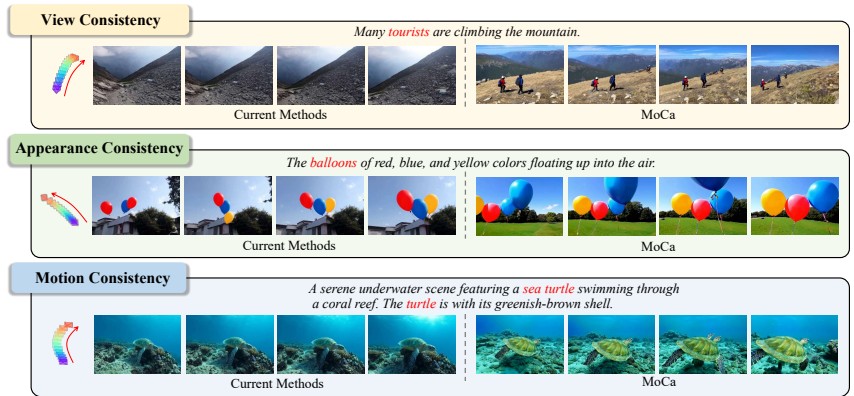

Figure 1: This figure outlines the core requirements for high-quality camera-controllable video generation: consistent object view, appearance, and motion. The foreground object should remain visible and retain its structure during camera movement, with stable texture and natural motion. Existing methods often fail to satisfy all three aspects simultaneously, whereas our approach demonstrates strong performance across all criteria.

## Abstract

Camera control is important in text-to-video generation for achieving realistic scene navigation and view synthesis. This control is defined by parameters that describe movement through 3D space, thereby introducing 3D consistency into the generation process. A core challenge for existing methods is achieving 3D consistency within the 2D pixel domain. Strategies that directly integrate camera conditions into text-to-video models often produce artifacts, while those relying on explicit 3D supervision face challenges with generalization. Both limitations originate from the gap between the 2D pixel space and the underlying 3D world. The key insight is that the projection of a smooth 3D camera movement produces consistency in object view, appearance, and motion across 2D frames. Inspired by this insight, we propose MoCa, a dual-branch framework that bridges this gap by modeling object consistency to implicitly learn 3D relationships between the camera and the scene. To ensure view consistency, we design a Spatial-Temporal Camera Encoder with Plücker embedding, which encodes camera trajectories into a geometrically grounded latent representation. For appearance consistency, we introduce a semantic guidance strategy that leverages persistent vision-language features to maintain object identity and texture across frames. To address motion consistency, we propose an object-aware motion disentanglement mechanism that separates object dynamics from global camera movement, ensuring precise camera control and natural object motion. Experiments show that MoCa achieves accurate camera control while preserving video quality, offering a practical and effective solution for camera-controllable video generation.

---

*Work done during an internship at Li Auto.

†Corresponding author.

## 1 INTRODUCTION

Recent years have witnessed the significant success of video generation models supported by the foundational diffusion model in content creation and movie production owing to their excellent multi-modal understanding and powerful generation capabilities, especially in text-conditioned generation (Blattmann et al., 2023; Chen et al., 2023; He et al., 2022; Singer et al., 2023; Zhou et al., 2022; Xie et al., 2025; Zhou et al., 2025). The growing demand for precise camera control to enhance video realism in applications like scene navigation and novel view synthesis remains unmet, as it requires models to understand the spatial relationship between the 2D pixel space and the 3D scene (Wang et al., 2025).

While a standard text-to-video model learns a mapping $f(\mathbf{P}) \rightarrow \mathbf{V}^{X \times Y \times T}$, where $X, Y$ denote pixel coordinates and $T$ represents the temporal dimension, its objective is to ensure alignment between the text prompt $\mathbf{P}$ and the video volume $\mathbf{V}$. Camera-controlled generation introduces a specific trajectory condition $\mathbf{C}$. This requires the model to understand the spatial relationships of objects from changing viewpoints, learning a more complex mapping $f(\mathbf{P}, \mathbf{C}) \rightarrow \mathbf{V}^{X \times Y \times Z \times T}$. Here, the $Z$-dimension represents the 3D spatial relation brought by the camera movement that must be consistently maintained. The challenge of learning this implicit 3D relation has led to different strategic approaches.

A common strategy is to treat camera control as an additional condition with standard 2D video generation frameworks. Methods such as MotionCtrl and CameraCtrl (Wang et al., 2024b; He et al., 2024) integrate camera parameters by temporal attention or element-wise addition in the Denoising U-Net. Without 3D spatial awareness, these approaches often struggle to maintain view and motion consistency, leading to artifacts like unnatural object dynamics. Another line of work seeks to explicitly learn the 3D relationship between the camera and objects with additional supervision. Techniques like VidCRAFT3 (Zheng et al., 2025) and ViewCrafter (Yu et al., 2024) convert video frames into 3D point clouds, while I2VControl-Camera (Feng et al., 2024) leverages RGB-D representations for better visual quality. These methods often rely on accurate 3D estimation, which limits their practicality and generalizability. Therefore, the gap between the 3D scene and 2D video pixels limits current methods in addressing the challenges of camera-controlled video generation.

The key insight for building an implicit bridge between 2D pixels and the 3D scene is recognizing that a camera-controlled video, as a 2D projection of a 3D scene, will demonstrate consistent object view, appearance, and motion across frames as a result of smooth camera movement. Accordingly, our method focuses on modeling three types of consistency to produce natural object coherence under camera control: View, Appearance, and Motion. **View Consistency:** The camera's position and orientation determine which objects appear in the frame. For text-to-video generation, objects described in the text prompt should be visible throughout the majority of the clip. **Appearance Consistency:** As the camera moves, the structure and texture of objects in the 2D projection should remain continuous and stable over time. **Motion Consistency:** When both the camera and objects move, the task becomes more difficult because the generated video reflects a combination of both motions. Camera movements change the field of view, primarily causing translation and scaling of static objects. Meanwhile, dynamic objects must not only follow the natural motion described in the text prompt, but also exhibit translation and scaling consistent with the camera's movement. Therefore, the three types of consistency, view, appearance, and motion that are observed in a 2D video imply a stable camera moving through a 3D scene. We believe objects should be consistent when the camera is controlled. We follow this insight to design our method.

In this paper, we propose a dual-branch fusion framework named MoCa, focusing on **M**odeling **O**bject consistency to enhance **Ca**mera-controlled video generation. To maintain view consistency, we adopt Plücker embedding and Spatial-Temporal Camera Encoder (ST-Encoder), which provides a geometrically interpretable representation that encodes camera trajectories at the pixel level in a latent space. To enhance appearance consistency, we propose a semantic guidance strategy that incorporates vision-language features from a pre-trained foundational model. The vision-language features serve as persistent global scene information to guide the fusion of camera-conditioned visual features, mitigating issues such as object distortion and texture collapse. As for motion consistency, we decouple video motion into camera movements and object dynamics. For precise pixel-level camera control, we adopt the ST-Encoder with Plücker embedding. Meanwhile, plausible object motion is also crucial for high-quality video generation. To achieve this, we propose

an object-aware motion disentanglement that separates object dynamics from global camera movements. Specifically, we extract the implicit structure and region information of foreground objects from the pre-trained foundational model, which serves as an object-aware mask to guide the motion disentanglement. This mechanism allows the model to maintain natural object dynamics while achieving precise camera control.

The main contributions of our work are as follows:

- We propose a method that learns view, appearance, and motion consistency to bridge 3D camera movement and its corresponding changes in 2D frames. This implicit learning of the scene-camera relationship results in enhanced camera-controlled video generation.

- We design a dual-branch framework comprising ReferenceNet and DenoisingNet, integrated with a semantic guidance strategy that injects visual-language features from the ReferenceNet to improve the appearance consistency of objects.

- We introduce an object-aware disentangling mechanism to separate object dynamics from camera movements, ensuring object motion is both faithful to the text prompt and consistent with the camera's movement.

## 2 RELATED WORK

### 2.1 TEXT-TO-VIDEO GENERATION

Text-to-video (T2V) generation is a challenging task that requires both high-fidelity visual realism and cross-modality consistency (Guo et al., 2024; Brooks et al., 2023; Wu et al., 2024; Ma et al., 2025; Menapace et al., 2024; Yang et al., 2022; Pan et al., 2025). In the early stages, research in video generation primarily relied on Generative Adversarial Networks (GANs) or Variational Autoencoders (VAEs) (Saito et al., 2017; Skorokhodov et al., 2022; Tulyakov et al., 2018; Vondrick et al., 2016). Despite the progress they made, the performance of these methods was still far from expectations. Recent attempts at text-to-video (T2V) generation mainly leverage diffusion models for their impressive capability and well-established open-sourced communities (Blattmann et al., 2023; Chen et al., 2023; Guo et al., 2024; He et al., 2022; Ho et al., 2022; Singer et al., 2023; Wang et al., 2023). As a pioneer in this field, some methods (Ho et al., 2022; He et al., 2022; Hong et al., 2022; Karras et al., 2023; Ruan et al., 2023) commonly employ video diffusion models (VDMs) that incorporate temporal convolutional and attention layers into the pre-trained image diffusion models. Follow-up works, VideoCrafter (Chen et al., 2023) and SVD (Blattmann et al., 2023) expand the application of video diffusion models to larger datasets, while TF-T2V (Wang et al., 2024a) directly learn from extensive text-free videos. Nonetheless, these methods encounter limitations in generating long videos, owing to the inherent constraints on capacity and scalability within the U-Net design. To overcome these constraints, DiT-based models (Brooks et al., 2024; Peebles & Xie, 2023) have emerged as a promising alternative, enabling direct generation of videos extending up to tens of seconds. Meanwhile, Sora (Zhang et al., 2023) adopts a unified visual representation, supporting large-scale training and synthesis of high-resolution videos exceeding one minute.

### 2.2 CAMERA-CONTROLLED VIDEO GENERATION

As a pioneering work, MotionCtrl (Wang et al., 2024b) learns camera control by conditioning pre-trained video models with extrinsic matrices. Follow-up works further improve the conditioning mechanisms. CameraCtrl (He et al., 2024) represents cameras as Plucker coordinates, which allows more stable ray-based rendering and view-dependent modeling. I2VControl-Camera (Feng et al., 2024) introduces point trajectory guidance for precise object-centric control. Building upon this, CamCo (Xu et al., 2024) integrates epipolar constraints into attention layers, and CamTrol (Hou et al., 2024) leverages explicit 3D point cloud representations. Related progress in explicit 3D modeling (Yang et al., 2025; Luo et al., 2024) demonstrates that enforcing multi-view geometric consistency through 3D representations can significantly improve viewpoint coherence, although these methods operate in the text-to-3D domain. Another line of works controls camera motion without training additional parameters. However, these methods often rely on additional guidance, such as depth or segmentation masks. Notably, these approaches adopt U-Net-based architectures

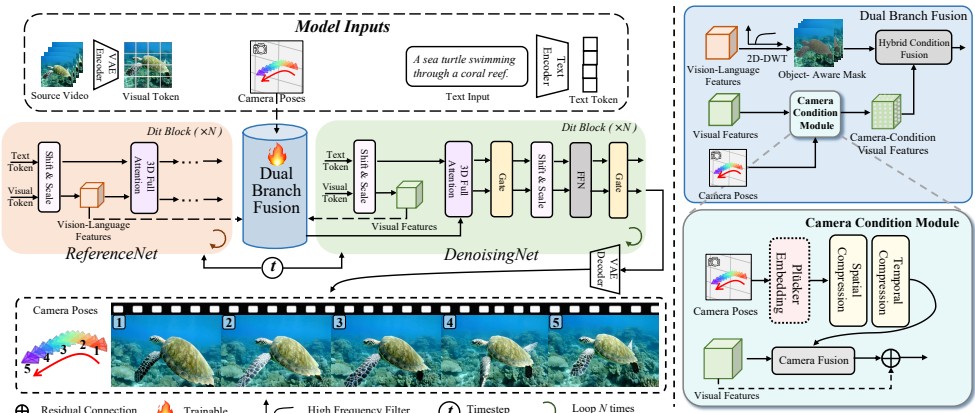

Figure 2: The overview of MoCa. To maintain view consistency, we utilize the Camera Condition Module with Plücker embedding to align camera rays with pixel-level visual representation. For appearance consistency, a semantic guidance strategy employs ReferenceNet's vision-language features to stabilize objects. Motion consistency is achieved by disentangling video motion into camera movement and object motion.

as their backbone. More recently, (Bahmani et al., 2024b;a; He et al., 2025) incorporates camera control into a video diffusion transformer architecture. Despite these advances, existing methods fail to generate dynamic content (e.g., object motion) under camera control without constructing specialized dynamic video datasets. Our work enhances dynamic content generation without the need for curating additional dynamic videos.

## 3 METHOD

Figure 2 illustrates the overall pipeline of MoCa. To model objects' consistency of view, appearance, and motion, we introduce a dual-branch fusion framework. For view consistency, we design a Spatial-Temporal Camera Encoder (ST-Encoder) with Plücker embedding as the primary form of camera parameters, aligning camera rays with pixel-level video representation. To enhance appearance consistency, we leverage vision-language alignment features from a foundational model and design a semantic guidance strategy to stabilize the appearance of objects. As for motion consistency, we propose an object-aware disentangling mechanism that guides the generation process to separate local object motion from camera movements.

### 3.1 CAMERA CONDITION MODULE

To ensure view consistency and prevent object misplacement, we introduce a camera condition module comprising a Camera Representation, Spatial-Temporal Camera Encoder, and Camera Fusion Module. This system encodes camera trajectories into pixel-level representations and integrates them into the denoising process, guaranteeing that objects remain aligned with the camera's viewpoint.

**Camera Representation.** For the camera representation, we adopt Plücker embedding (Sitzmann et al., 2021) following recent works (He et al., 2024; Bahmani et al., 2024a), which provides strong geometric interpretation and fine-grained camera information. Specifically, given camera extrinsic matrix $\mathbf{E} = [\mathbf{R}; \mathbf{t}] \in \mathbb{R}^{3 \times 4}$ and intrinsic matrix $\mathbf{K} \in \mathbb{R}^{3 \times 3}$, we compute the Plücker embedding $\mathbf{p} = (\mathbf{o} \times \mathbf{d}', \mathbf{d}')$ for each pixel $(u, v)$. Here, $\mathbf{o}$ represents the camera center in world coordinates, the ray direction from camera to pixel is defined as $\mathbf{d} = \mathbf{R}\mathbf{K}^{-1}[u, v, 1]^T + \mathbf{t}$, and $\mathbf{d}'$ is the normalized $\mathbf{d}$. The final Plücker embedding $\mathbf{P}_i \in \mathbb{R}^{6 \times h \times w}$ is constructed for each frame, where $h$ and $w$ are the height and width of the frame.

**Spatial-Temporal Camera Encoder.** To integrate Plücker embedding into the generation process, we design a Spatial-Temporal Camera Encoder that transforms camera conditions into latent representations that are both spatially and temporally consistent with visual latents. In the spatial domain, a progressive convolutional architecture with downsampling, convolutional, and residual blocks ex-

tracts pixel-level spatial features for camera motion. To incorporate temporal dynamics, we introduce dedicated temporal convolutions across the frame sequence. The resulting spatial-temporal camera representations are fused with the visual features within the denoising process.

**Camera Fusion Module.** To effectively integrate camera representations with visual features, we adopted the fusion strategy from existing methods (Bahmani et al., 2024a; He et al., 2025) that injects control signals into each diffusion transformer (DiT) block. Specifically, we apply a cross-attention mechanism in each DiT block to fuse camera representations. This design allows the model to dynamically modulate visual features based on spatial-temporal camera conditions, enabling fine-grained and precise controllability. By injecting geometry-aware camera representations into the generation process, the model maintains alignment between camera viewpoints and semantic objects, reducing cases where the described foreground fails to appear. Additionally, we explore alternative fusion strategies for camera signals, which are discussed in the ablation study.

## 3.2 SEMANTIC GUIDANCE STRATEGY

Appearance, such as the texture and scale of the object, is critical in video generation, which directly affects the temporal consistency and visual integrity. Existing models suffer from object distortion or texture collapse in complex dynamic scenes, especially under intense camera movements. We trace this limitation to the additional camera signals, which weaken the generative power of the base model. To address this, we propose a semantic guidance strategy that injects the visual-language features from a frozen foundational model into the generation process.

Specifically, we extract the vision-language features from the visual branch in each DiT block of the ReferenceNet, and inject them into each DiT block of the DenoisingNet. We consider that these features are aligned in both visual and semantic spaces, serving as stable appearance guidance for the whole scene. Therefore, our semantic guidance strategy reinforces vision-language alignment and enhances the object appearance consistency of generated videos. Previous research (Tian et al., 2024; Hu, 2024; Ling et al., 2025) has demonstrated the influence of utilizing analogous structures in maintaining the consistency of the object's identity. In our setup, ReferenceNet shares the same structure as DenoisingNet. Both the ReferenceNet and the DenoisingNet are initialized with weights inherited from the original pretrained DiT architecture.

## 3.3 MOTION DISENTANGLEMENT

In this section, we present a motion disentangling mechanism for separating object motion from camera movements, which is designed to enhance motion consistency. We decompose the overall video motion into camera movement and object motion. Given that camera control has been addressed in Section 3.1, this section focuses on our approach to modeling natural object motion. Specifically, we address the entanglement of object motion and camera motion by leveraging frequency-domain analysis. We utilize a multi-level 2D Discrete Wavelet Transform (2D-DWT) (Shahbahrami, 2012; Huang et al., 2005; Mushtaq et al., 2015) to extract high-frequency components from visual features, highlighting the structures and regions of objects. These high-frequency components guide the model to focus on natural object motion while improving precise camera controllability.

Current methods often fail to balance global camera movement and local object motion. When strong camera motion is applied, objects remain completely still, failing to show natural dynamics such as a fish swimming or a person walking. This limitation arises because video diffusion models entangle object and camera motion, making it difficult to maintain independent object dynamics.

**High-Frequency Object-Aware Masking.** To improve the realism of object motion, we propose an implicit object masking strategy that guides the model to separate local object motion from global camera movements. Specifically, we leverage the vision-language features from the foundational model to extract visual information that highlights foreground object structures and regions. Inspired by frequency-domain analysis in image processing, we apply a multi-level 2D Discrete Wavelet Transform (2D-DWT) (Shahbahrami, 2012; Huang et al., 2005; Mushtaq et al., 2015) to vision-language features across different orientations, capturing localized spatial-frequency information. This allows us to retain fine-grained structural cues that are critical for the precise localization of

| Methods | FID ↓ | FVD ↓ | CLIPSIM ↑ | TransErr ↓ | RotErr ↓ | OC ↑ | BC ↑ | MS ↑ |
|---|---|---|---|---|---|---|---|---|
| **RealEstate10K** | | | | | | | | |
| MotionCtrl(Animatediff-Based) (Wang et al., 2024b) | 246.6 | 870.8 | 0.309 | 0.716 | 0.213 | 94.6% | 95.8% | 97.8% |
| CameraCtrl (He et al., 2024) | 255.8 | 931.5 | 0.305 | 0.708 | 0.204 | 94.3% | 94.7% | 97.7% |
| AC3D (Bahmani et al., 2024a) | 225.2 | 683.4 | 0.309 | **0.695** | **0.196** | **95.1%** | 95.3% | **98.5%** |
| Ours | **207.4** | **667.9** | **0.312** | 0.703 | 0.208 | 94.9% | **96.4%** | **98.5%** |
| **VidGen** | | | | | | | | |
| MotionCtrl(Animatediff-Based) (Wang et al., 2024b) | 274.0 | 1858.2 | 0.333 | **0.722** | 0.107 | 92.6% | 93.2% | 97.1% |
| CameraCtrl (He et al., 2024) | 266.3 | 1905.1 | 0.339 | 0.731 | 0.089 | 92.9% | 93.1% | 96.9% |
| AC3D (Bahmani et al., 2024a) | **228.4** | 1712.0 | 0.345 | 0.727 | 0.084 | 93.5% | 94.7% | 97.7% |
| Ours | 232.2 | **1643.7** | **0.349** | 0.724 | **0.081** | **94.7%** | **95.1%** | **98.3%** |

Table 1: Quantitative comparison on RealEstate10K and VidGen datasets. Lower is better (↓), higher is better (↑). **Bold** indicates top-1 performance.

object regions and effective motion disentanglement. The visualization of 2D-DWT is discussed in the appendix.

**Hybrid Condition Fusion.** We propose a hybrid conditioning fusion to fuse the object-aware mask with camera-conditioned visual features. It strategically employs cross-attention for spatial conditioning fusion, with the temporal attention explicitly enforcing inter-frame consistency. The fusion strategy allows the model to dynamically modulate object-aware guidance and camera-conditioned features, enforcing accurate camera movements and preserving natural object motion. Meanwhile, this strategy enables the model to maintain motion consistency across frames. Powered by the structure and localization information from the foundational model, our method enhances dual motion disentanglement and separates object motion from the camera movements.

## 4 EXPERIMENTS

### 4.1 IMPLEMENTATION DETAILS

**Datasets.** During training, our model is fine-tuned from CogVideoX (Yang et al., 2024) on RealEstate10K (Zhou et al., 2018), which has around 65K video clips with per-frame camera parameters (extrinsics and intrinsics). Moreover, we employ random interval sampling of video frames to enhance the model's capability in handling complex camera inputs. For evaluation, we assess performance on both RealEstate10K and the VidGen dataset. VidGen (Tan et al., 2024) consists of a large collection of text-video pairs, primarily featuring dynamic scenes. In contrast, RealEstate10K focuses on static scenes, showcasing furnishings in indoor settings and natural landscapes in outdoor environments. We leverage this dataset to validate the effectiveness of our approach in object stability and motion consistency, thereby demonstrating its generalization to complex, dynamic scenes.

**Metrics.** We evaluate performance using a comprehensive set of quantitative metrics. For common evaluation, we report FID, FVD, and CLIPSIM scores. To evaluate camera accuracy, we follow CameraCtrl (He et al., 2024), using rotation and normalized translation errors from Mega-SAM (Li et al., 2024) reconstructed trajectories. To further assess foreground object consistency and background consistency, we adopt Object Consistency (OC), Background Consistency (BC), and Motion Smoothness(MS) scores from VBench (Huang et al., 2024), respectively. We consider OC, BC, and MS scores from VBench as standard and widely used evaluation metrics for text-to-video generation. Moreover, these metrics collectively capture the different aspects of consistency that our method aims to improve.

### 4.2 QUANTITATIVE COMPARISON

To evaluate the effectiveness of our method, we compare it with existing methods on Realestate10K and VidGen. For a fair comparison, all videos are uniformly downsampled to 16 frames and cropped to the same size. It demonstrates that our method achieves superior visual quality on RealEstate10K compared to existing methods, while maintaining competitive performance in camera controllability. These results validate our method's effectiveness for static scene camera controllability, particularly in achieving enhanced realism and stability beyond existing methods. Moreover, we conduct extensive experiments on the dynamic scene dataset VidGen. Our method outperforms previous methods across key metrics, such as CLIPSIM and OC scores. In addition to achieving superior performance, our method achieves suboptimal performance on metrics such as FID and TransErr. Owing

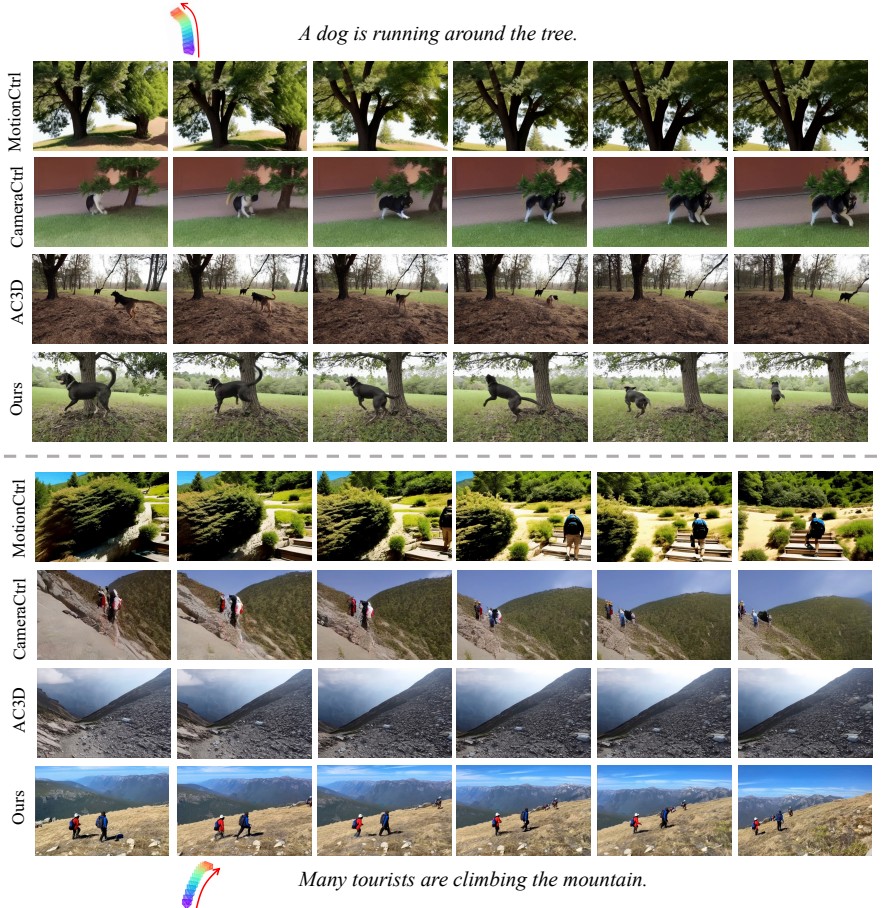

Figure 3: This figure presents a qualitative comparison between our method and existing approaches.

to the disentanglement of the object motion from the camera movement, our method achieves stable performance in motion smoothness, even in the dynamic scene, unlike other methods that exhibit significant degradation. Experiments show that our method achieves high-quality generation with view, appearance, and motion consistency.

### 4.3 QUALITATIVE COMPARISON

We present qualitative comparisons in Figure 3, where the sequence from left to right represents the start to the end of the video. For the upper example, MotionCtrl fails to maintain view consistency, with no dog appearing across all frames. CameraCtrl effectively preserves view consistency but fails to maintain the appearance consistency of the dog, showing texture artifacts. AC3D exhibits unnatural motion and similarly fails to preserve the appearance consistency of the dog. In contrast, our method achieves excellent consistency in view, appearance, and motion.

For the lower example, no tourists appear in all frames of AC3D, which clearly violates view consistency. CameraCtrl depicts the tourists' appearance very blurrily. The tourists' outlines in MotionCtrl are relatively accurate, but the motion continuity is poor, and they don't appear in the first few frames, violating view consistency. Our method performs much better overall. From the qualitative results, existing methods struggle to effectively maintain view, appearance, and motion consistency in the frames. The fundamental reason is still the lack of understanding of 3D space. Our method achieves this through constraints on objects.

### 4.4 QUALITATIVE RESULTS OF MOCA UNDER CONFLICTING MOTION

To evaluate the effectiveness of motion disentanglement, we further test MoCa in scenarios where the object motion direction described in the text conflicts with the camera input. For example, given

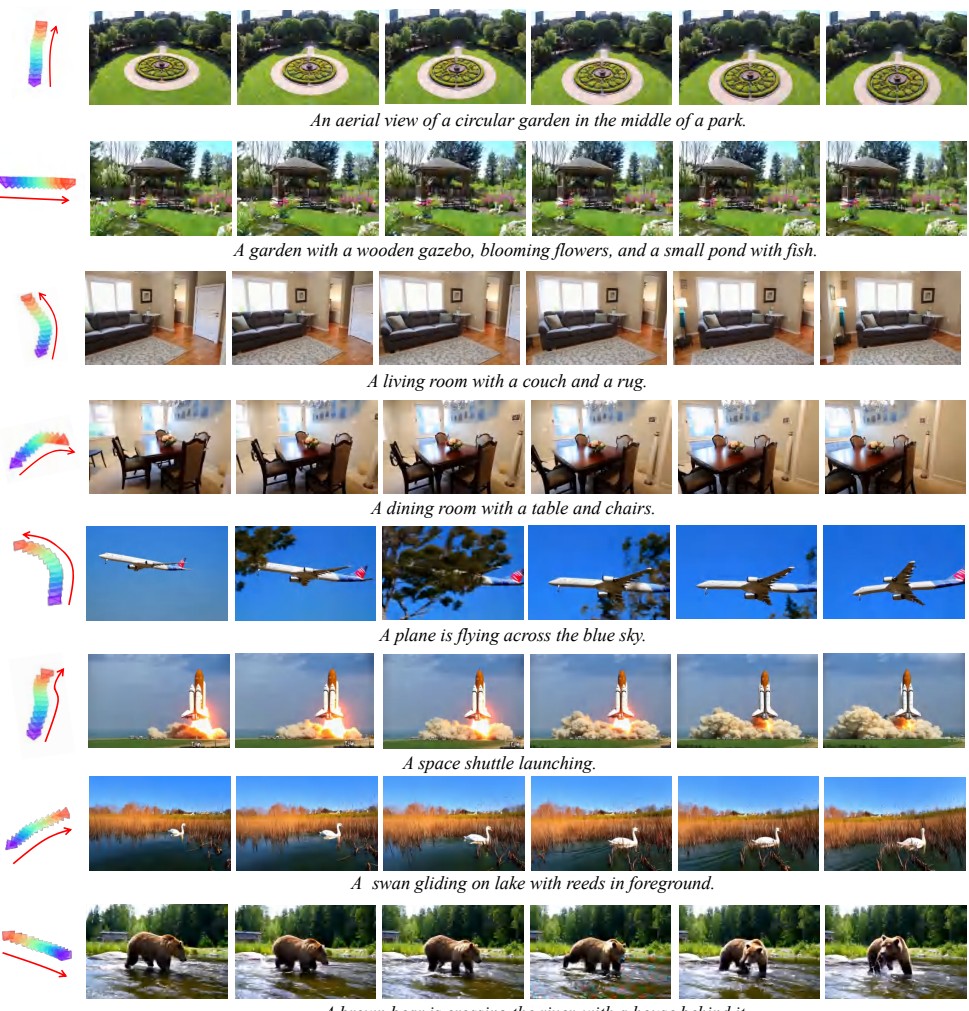

*An aerial view of a circular garden in the middle of a park.*

*A garden with a wooden gazebo, blooming flowers, and a small pond with fish.*

*A living room with a couch and a rug.*

*A dining room with a table and chairs.*

*A plane is flying across the blue sky.*

*A space shuttle launching.*

*A swan gliding on lake with reeds in foreground.*

*A brown bear is crossing the river, with a house behind it.*

Figure 4: Qualitative results of our method in both static and dynamic scenes.

the prompt "a bird flying from right to left" while the camera pans to the right, MoCa generates a bird moving against the camera motion correctly. As shown in Figure 5, MoCa ensures that the object motion (guided by the text input) is not overridden or distorted by the camera movement (guided by the camera input). This result indicates that our motion disentanglement mechanism effectively decouples object motion from camera movements.

To further validate this, we compare our model with other existing methods under the same text and camera inputs. We observe that they often fail to realize the text-specified object motion direction, as their object motion remains entangled with the camera motion. This contrast further confirms the necessity and effectiveness of our motion disentanglement strategy.

### 4.5 QUALITATIVE RESULTS OF MoCa ACROSS DIVERSE SCENARIOS

In this section, we present more results of our MoCa, especially on tasks of varying difficulty. As shown in Figure 4, there are eight examples from top to bottom, divided into four different types with increasing difficulty. The first group is outdoor scenes where most objects are in the distance, and there are no complex spatial relationships. The second group is indoor scenes, where objects have clearer 3D relationships, such as a table in the center of the picture and a wall behind the table. Both groups of examples require the model to maintain the view and appearance consistency for a good visual effect. The following two groups of examples are more complex, describing the results of a dynamic object under camera movement. In the third group, airplanes and rockets move over a large area in the scene, and there are even complex occlusions. For example, trees block

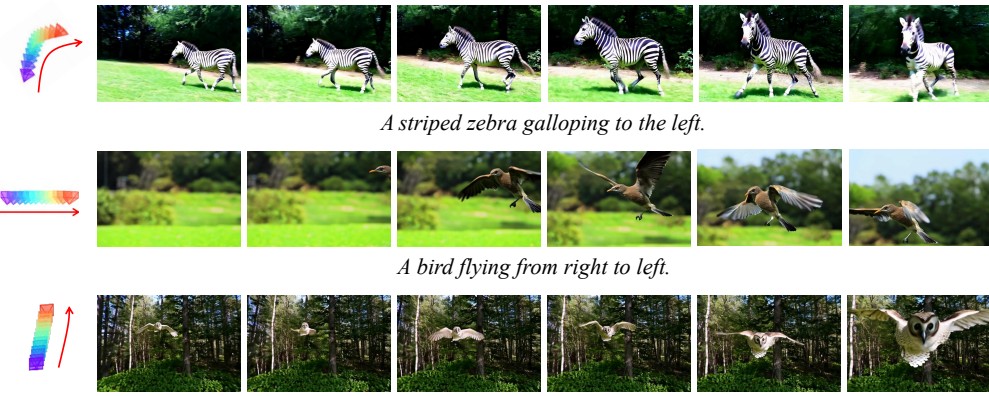

*A striped zebra galloping to the left.*

*A bird flying from right to left.*

*An owl is gliding from far to near.*

Figure 5: Qualitative results of our method under the conflicting motion. It shows that our motion disentanglement strategy decouples object motion from camera movements effectively. The foreground object motion is not overridden or distorted by the camera input.

the airplane, and smoke blocks the tail flame. Due to the understanding of 3D space, our method can handle these scenes well and achieve good view, appearance, and motion consistency. For the fourth group of examples, the objects are animals that have complex self-initiated movements, which require the model not only to understand camera movement, but also to decouple the animals' own dynamics. Due to our motion decoupling mechanism, we can handle these situations well. In summary, our method achieves good performance in camera-controlled video generation, especially in view, appearance, and motion consistency.

## 4.6 ABLATION STUDIES

**Plücker Embedding.** We directly use the numerical values of camera parameters (extrinsics and intrinsics) to evaluate the contribution of Plücker embedding. We utilize the linear projection to encode the camera parameters along the spatial-temporal dimension of the visual features in the generation process. The experimental results are illustrated in Table 2. We find that using the Plücker embedding as the camera representation yields more precise camera control. Meanwhile, due to the strong geometric interpretation, the powerful geometric interpretation directly results in superior consistency for both objects and the background. Incorporating original numerical values directly could compromise geometric relationships.

**Semantic Guidance Strategy.** To assess the role of vision-language features from the foundational model, we conduct both qualitative and quantitative studies. As shown in Table 2 and Figure 6, the introduction of the semantic guidance strategy leads to notable enhancements in object appearance consistency. Under strong camera movements, objects maintain their appearance without distortion. For example, without the semantic guidance strategy, the sea turtle in Figure 6 exhibits significant geometric distortion. Furthermore, the improved preservation of object appearance enables more effective high-frequency decomposition in subsequent processing stages.

**High-Frequency Object Masking.** As discussed in Sec. 3.3, we extract an object-aware mask through high-frequency decomposition to achieve motion disentanglement. We recognize the mask as the fine-grained cues of the region and localization of the foreground object. To validate its importance, we perform an ablation by removing this decomposition and directly fusing camera-conditioned visual features with vision-language features. The result in Table 2 shows that without high-frequency masking, both object and background consistency scores drop considerably. The mask helps the model better identify objects in the video frame by emphasizing foreground structures and localization, achieving motion disentanglement. Meanwhile, our motion disentanglement leads to a marked increase in motion smoothness, particularly in dynamic scenes.

**Camera Fusion Strategies.** Regarding camera fusion strategies, we evaluated two distinct approaches: element-wise addition fusion and cross-attention fusion. Existing approaches, such as CameraCtrl (He et al., 2024) and AC3D (Bahmani et al., 2024a), typically fuse camera conditions via element-wise addition. Specifically, they directly fuse image latent features and camera pose fea-

| Methods | FID ↓ | FVD ↓ | CLIPSIM ↑ | TransErr ↓ | RotErr ↓ | OC ↑ | BC ↑ | MS ↑ |
|---------|-------|-------|-----------|------------|----------|------|------|------|
| **RealEstate10K** | | | | | | | | |
| W/O PLÜCKER EMBEDDING | 225.8 | 694.7 | 0.309 | 0.758 | 0.210 | 93.5% | 95.1% | 98.4% |
| W/O SEMANTIC GUIDANCE | 243.1 | 705.6 | 0.308 | 0.722 | **0.198** | 94.1% | 95.8% | 97.9% |
| W/O HIGH-FREQUENCY MODELING | 235.4 | **649.8** | 0.309 | 0.744 | 0.209 | 94.5% | 94.9% | 98.0% |
| OURS (FULL, ADDITION FUSION) | 236.2 | 771.8 | 0.310 | 0.738 | 0.211 | 94.6% | 95.1% | 98.2% |
| OURS (FULL, ATTENTION FUSION) | **207.4** | 667.9 | **0.312** | **0.703** | 0.208 | **94.9%** | **96.4%** | **98.5%** |
| **VidGen** | | | | | | | | |
| W/O PLÜCKER EMBEDDING | 258.5 | 1716.4 | 0.340 | 0.747 | 0.109 | 93.2% | 94.5% | 97.6% |
| W/O SEMANTIC GUIDANCE | **231.7** | 1739.8 | 0.336 | **0.723** | 0.096 | 93.4% | 94.0% | 95.2% |
| W/O HIGH-FREQUENCY MODELING | 233.4 | 1735.4 | 0.339 | 0.733 | 0.092 | 94.3% | 94.6% | 97.4% |
| OURS (ADDITION FUSION) | 248.1 | 1738.6 | 0.345 | 0.732 | 0.084 | **94.9%** | 94.7% | 97.4% |
| OURS (ATTENTION FUSION) | 232.2 | **1643.7** | **0.349** | 0.724 | **0.081** | 94.7% | **95.1%** | **98.3%** |

Table 2: Ablation studies on RealEstate10K and VidGen datasets. Lower is better (↓), higher is better (↑). **Bold** indicates top-1 performance.

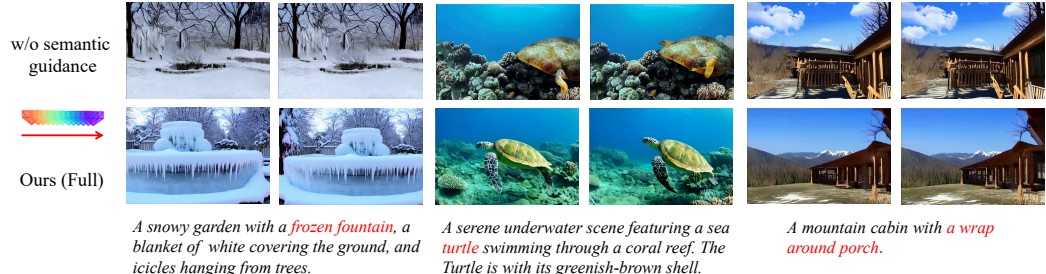

*A snowy garden with a frozen fountain, a blanket of white covering the ground, and icicles hanging from trees.*

*A serene underwater scene featuring a sea turtle swimming through a coral reef. The Turtle is with its greenish-brown shell.*

*A mountain cabin with a wrap around porch.*

Figure 6: It shows our ablation study on the semantic guidance strategy. Without it, we observe that the generated videos suffer from object distortions.

tures through pixel-wise addition. However, as shown in Table 2, this fusion strategy falls short in achieving accurate camera control due to its limited capacity for pixel-wise understanding between camera parameters and visual features. In contrast, our adopted camera fusion strategy helps achieve better performance on TransErr and RotErr. Benefiting from its stronger pixel-level understanding, the attention fusion also outperforms addition fusion on metrics such as FID, FVD, and CLIPSIM, leading to more realistic and semantically consistent video generation.

### 4.7 ANALYSIS OF MODEL COMPLEXITY AND RUNTIME OVERHEAD.

In this section, we measure per-sample latency on a single H200 GPU using BF16 precision and 50 denoising steps. MoCa has an inference latency of 291.2s. The dual-branch fusion introduces 74.6s of overhead, and the 2D-DWT and hybrid condition fusion add 9.73s (≈3.3% of the total time). Although MoCa has a higher inference latency than the backbone, it produces better visual-quality and camera-controllable videos. In practical video generation, users are typically more concerned with visual quality and controllability than with inference speed. For the latest diffusion-based video generation models, generating a few seconds of a satisfying video requires several minutes of computation. This inherent mismatch between video length and inference time highlights that the video quality is more important than gains in inference latency.

## 5 CONCLUSION

This work introduces MoCa, a framework for camera-controllable video generation that addresses the challenge of 3D consistency in 2D pixel space. By modeling object consistency across view, appearance, and motion, MoCa bridges the gap between 2D pixel space and the underlying 3D scene. Our approach incorporates a camera condition module to maintain view consistency, a semantic guidance strategy to preserve object appearance, and an object-aware motion disentanglement mechanism to separate local object motion from global camera movements. Experimental results demonstrate that MoCa achieves accurate camera control while maintaining high video quality, offering a practical solution for consistent and controllable video generation.

## 6 ACKNOWLEDGMENTS

This work was supported by the National Natural Science Foundation of China (NSFC) grant U22A2094.

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

## A    APPENDIX / SUPPLEMENTAL MATERIAL

This supplementary document offers additional details, extended analyses, and further experimental results that support the main content of the paper. It includes implementation specifics, extra qualitative comparisons, and more visualizations of the generated videos. We structure the appendix as follows: Appendix B claims the usage of large language models. Appendix C provides a brief review of ReferenceNet-based camera-controlled video generation. Appendix D explains the use of the 2D Discrete Wavelet Transform for high-frequency object-aware masking. Appendix E presents additional experiment details, including settings and dataset information. Appendix F includes more qualitative results, especially in dynamic scenes. Appendix G discusses the limitations.

## B    THE USAGE OF LARGE LANGUAGE MODELS

We acknowledge the use of a large language model for limited assistance in the writing of this paper. The tool was used exclusively for proofreading and improving the linguistic fluency of the text. All scientific content, including the research ideas, methodology, experiment, and analysis, was conducted solely by the authors.

## C    REFERENCENET BASED CAMERA-CONTROLLED VIDEO GENERATION

### C.1    TEXT-TO-VIDEO DIFFUSION MODELS.

Modern text-to-video (T2V) models typically build upon a pre-trained text-to-image (T2I) model, such as Stable Diffusion (SD), a pioneering framework that follows the design of the Latent Diffusion Model (LDM). In this work, we follow the T2V model definition from CameraCtrl. As an essential part of SD, Variational AutoEncoder compresses the feature distribution of the original image, denoted as $x_0$, into a latent space representation $z_0$. The encoding operation extracts the image essence as $z_0 = E(x_0)$, whereas the decoding counterpart reconstructs the image from the latent via $x_0 = D(z_0)$. The diffusion process is then conducted in the latent space, which significantly reduces computational overhead without compromising generation performance.

During the diffusion phase, these models usually corrupt the latent $z_0$ by adding Gaussian noise $\epsilon$ according to a predefined schedule inherited from Denoising Diffusion Probabilistic Model (DDPM) or its deterministic variant DDIM. Then the denoising network is optimized to reverse this process by progressively eliminating the introduced noise directed by some conditional embeddings $c$, to yield videos that adhere to the prescribed text prompts. The training objective can be formulated as follows:

$$\mathcal{L} = E_{t,\epsilon,\mathbf{z}_0^{1:N},\mathbf{c}} \left[ \left\| \epsilon - \epsilon_\theta(\mathbf{z}_t^{1:N}, t, \mathbf{c}) \right\|^2 \right]$$

where N denotes the number of video frames, $c$ is the text embeddings transformed from the input prompts utilizing the CLIP ViT-L/14 text encoder, $\epsilon_\theta$ is a Unet with learnable weights $\theta$. The Unet is composed of pairs of down/up blocks as well as a middle block. Each block consists of ResNets, spatial and temporal self-attention layers, together with cross-attention layers that interact with text conditions, thereby promoting the model's capability to generate videos that are semantically consistent with text.

### C.2    CAMERA-CONTROLLED TEXT-TO-VIDEO GENERATION.

In the field of text-to-video generation, tasks about adding camera trajectory control $s_t$ have already been extensively explored. By incorporating structural camera signals, the video generation results can be more controllable. Specifically, the camera trajectories are first processed by a special encoder $\phi(\cdot)$ and then fed into the video generator for further operations. Consequently, the objective of the generator with guidance from camera signals can be formulated as follows:

$$\mathcal{L} = E_{t,\epsilon,\mathbf{z}_0^{1:N},\mathbf{c},s_t} \left[ \left\| \epsilon - \epsilon_\theta(\mathbf{z}_t^{1:N}, t, \mathbf{c}, \phi(s_t)) \right\|^2 \right]$$

where $N$ is the number of video frames, $c$ is the text embeddings, and $s_t$ means camera trajectories at different timesteps.

## C.3 REFERENCENET.

Prior research has pointed out that utilizing analogous structures is crucial in maintaining the identity consistency of the target object. Therefore, ReferenceNet, which mirrors the architectural design of SD and operates in parallel with the Denoising Unet, is widely adopted to assist in modelling complex image or video features. EMO and EchoMimic facilitate the self-attention mechanism in the ReferenceNet to extract reference image features into the attention layers of the corresponding block in the Denoising U-Net, making the facial identity more consistent throughout the video. Hallo integrates features from the same spatial resolution layers into the Denoising Unet to enhance the visual texture information of both portraits and backgrounds in the generated videos. Meanwhile, as ReferenceNet shares an identical network structure and initialization weights, the Denoising Unet can selectively learn some correlated features from it in the same feature space. AnimateAnyone leverages the learned reference image features from ReferenceNet to produce a well-initialized latent, thus accelerating the entire network training process of the Denoising U-Net.

## D 2D-DWT IN HIGH-FREQUENCY OBJECT-AWARE MASKING

As mentioned in Sec.3.3, we use a high-frequency object masking strategy to extract an object-aware mask for better separating objects and background. In this section, we give more details about this strategy. Formally, given the vision-language features from the foundational model $\mathbf{X} \in \mathbb{R}^{B \times C \times H \times W}$, we decompose them into four frequency sub-bands using multi-level 2D-DWT:

$$\text{DWT}(\mathbf{X}) \rightarrow \{\mathbf{LL}, \mathbf{LH}, \mathbf{HL}, \mathbf{HH}\}, \tag{1}$$

where $\mathbf{LL}$ denotes the low-frequency approximation coefficients, and $\mathbf{LH}$, $\mathbf{HL}$, $\mathbf{HH}$ represent high-frequency details in horizontal, vertical, and diagonal directions, respectively. To obtain these components, we adopt separable Haar wavelet filters and perform sequential convolutions:

$$h_L = \frac{1}{\sqrt{2}}[1, 1], \quad h_H = \frac{1}{\sqrt{2}}[-1, 1], \tag{2}$$

$$\begin{aligned}
\mathbf{LL} &= (\mathbf{X} * h_L^\top) * h_L, \quad \mathbf{LH} = (\mathbf{X} * h_L^\top) * h_H, \\
\mathbf{HL} &= (\mathbf{X} * h_H^\top) * h_L, \quad \mathbf{HH} = (\mathbf{X} * h_H^\top) * h_H.
\end{aligned} \tag{3}$$

Unlike conventional temporal wavelet transforms or global 2D Fourier transforms, our multi-level 2D-DWT preserves localized spatial-frequency characteristics, which are essential for accurately identifying object contours in dynamic scenes. To emphasize fine-grained structural cues, we discard the low-frequency component $\mathbf{LL}$ and retain only the high-frequency sub-bands. These are then used to reconstruct a high-frequency-enhanced representation via inverse DWT (iDWT):

$$\mathbf{X}_{\text{high}} = \text{iDWT}(0, \mathbf{LH}, \mathbf{HL}, \mathbf{HH}). \tag{4}$$

As visualized in Fig. 7, this operation highlights the structure and localization of foreground objects in the latent space. We view the high-frequency representation as an object-aware mask. The mask enhances motion consistency performance by improving the model's object identification capability. It achieves this by emphasizing foreground structures in the latent space, which promotes a clearer separation between foreground and background elements.

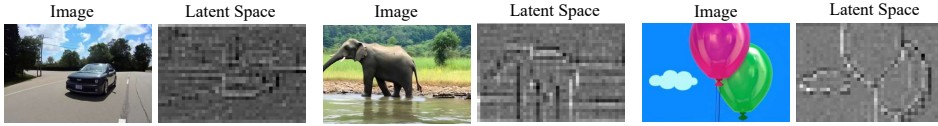

Figure 7: This figure presents a visualization of applying high-frequency decomposition to visual features in a latent space. Our strategy yields an object-aware mask that effectively captures the structure and localization of the foreground object. As shown in the right case, our high-frequency decomposition can accurately extract the structure of all objects, even in scenes containing multiple objects of different classes.

# E    MORE EXPERIMENT DETAILS

## E.1    TRAINING AND INFERENCE DETAILS

Our model is built upon CogVideoX (Yang et al., 2024), a transformer-based text-to-video diffusion model, which demonstrates strong performance on both automated metrics and human evaluations. Similar to ReferenceNet, the weights of our DenoisingNet are inherited from the original CogVideoX and remain frozen during training. Only the dual-branch fusion module is optimized. We employ the AdamW optimizer, an initial learning rate of $1 \times 10^{-4}$, epsilon set to $1 \times 10^{-8}$, weight decay of $1 \times 10^{-4}$, and beta values of 0.9 and 0.95.

During inference, to increase the magnitude of camera motion and enhance the challenge of camera-controllable video generation, we sample the input camera trajectory by selecting 49 frames at equal intervals from the original 98-frame sequence.

## E.2    DATASET

During training, our model is fine-tuned from CogVideoX on the RealEstate10K dataset, which comprises approximately 65K video clips extracted from around 10,000 YouTube videos, totaling about 10 million frames. For each clip, camera parameters (extrinsics and intrinsics) are provided for every frame, forming a continuous trajectory estimated by running SLAM and bundle adjustment algorithms on the original videos. This dataset supports view synthesis and 3D computer vision tasks, with scenes often focusing on static environments such as furnished indoor spaces and natural outdoor landscapes. Meanwhile, we employ random interval sampling of video frames to enhance the model's capability in handling complex camera inputs.

For evaluation, we test performance on both RealEstate10K and the VidGen dataset. VidGen is a large-scale collection of text-video pairs designed for text-to-video generation. The dataset is created through a meticulous curation process involving rough and fine-grained filtering to ensure high-quality videos with detailed, temporally consistent captions. In contrast to the static scenes in RealEstate10K, VidGen features predominantly dynamic scenes. Figure 8 shows samples of scenes from the two datasets.

We leverage the VidGen dataset to validate the effectiveness of our approach in terms of object stability and motion consistency. The diverse and complex dynamics within VidGen's videos enable a rigorous test of our model's ability to generalize beyond static environments, thereby demonstrating its robustness in handling complex, dynamic scenes, as visualized in the qualitative results shown in Figure 11, 12.

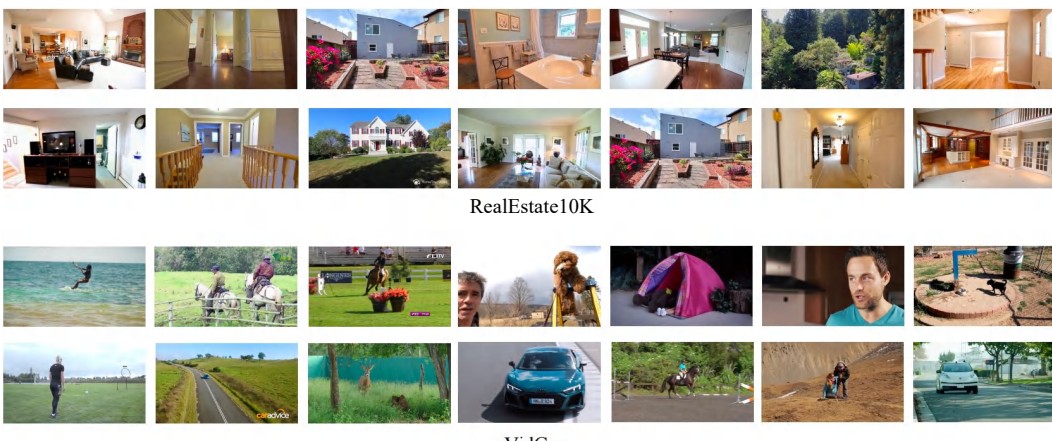

RealEstate10K

VidGen

Figure 8: The overview of the dataset RealEstate10K and VidGen.

# F MORE EXPERIMENT RESULTS

## F.1 QUALITATIVE COMPARISON ON DYNAMIC SCENES

In this section, we present additional qualitative comparison results. The sequence from left to right in the figure represents the progression from the start to the end of the video. As shown in Figure 9, we provide three new challenging cases, where each case involves animals or objects with self-driven motion. Our method demonstrates excellent performance in terms of view, appearance, and motion consistency.

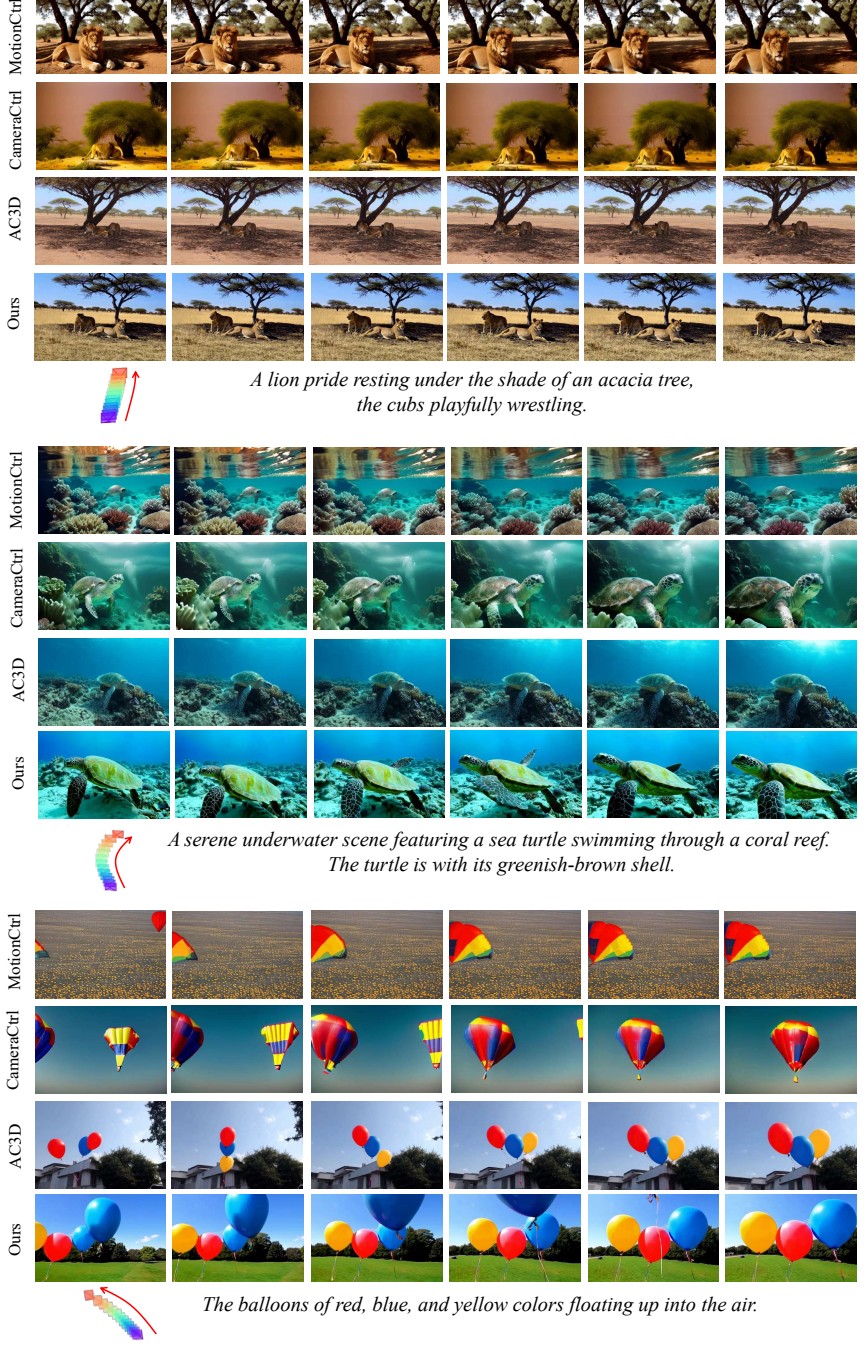

Figure 9: More qualitative comparison between our method and existing approaches.

## F.2 QUALITATIVE COMPARISON ON STATIC SCENES

In this section, we present additional qualitative comparison results. The sequence from left to right in the figure represents the progression from the start to the end of the video. As shown in Figure 10, we provide three new cases of static scenes. Our method demonstrates excellent performance in terms of view, appearance, and motion consistency.

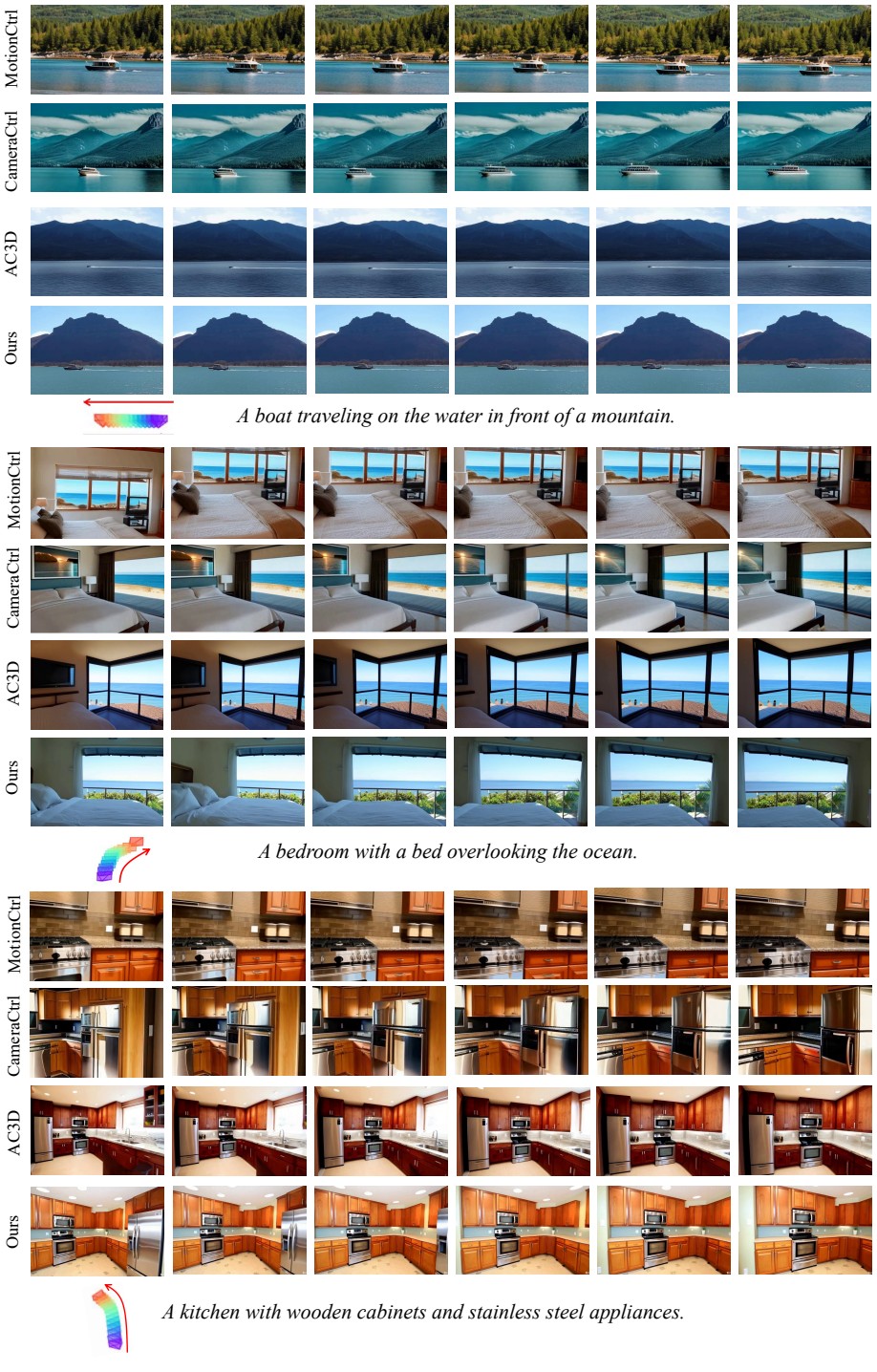

Figure 10: More qualitative comparison between our method and existing approaches.

## F.3 More Qualitative Results on Dynamic Scenes

In this section, we provide additional qualitative results under dynamic scene settings. Dynamic scenes are characterized by the presence of distinct objects with self-driven motion and a clear separation between foreground and background elements, presenting a particularly challenging scenario for video generation models.

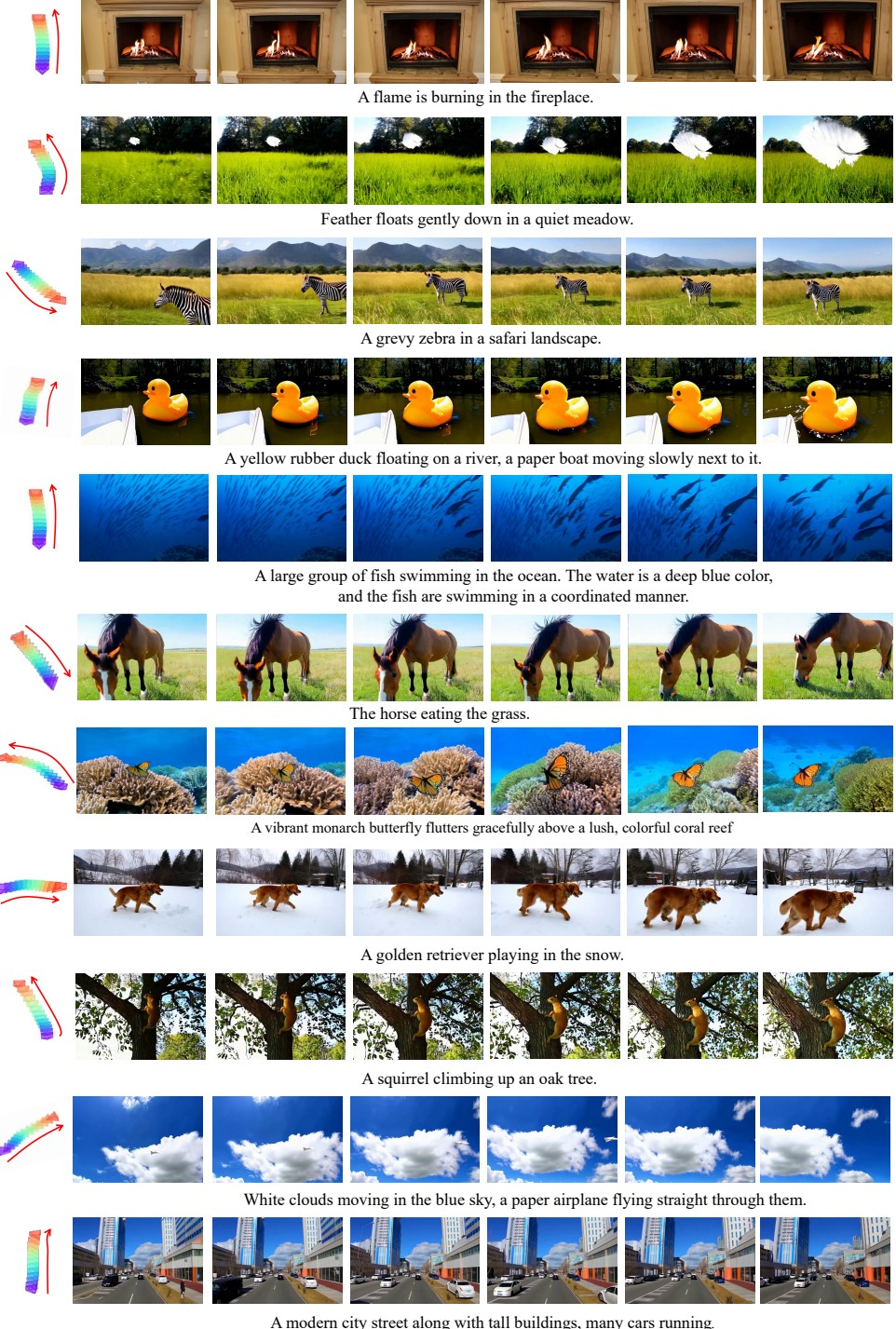

Figure 11: More qualitative results of MoCa on dynamic scenes.

## F.4 MORE QUALITATIVE RESULTS ON STATIC SCENES

In this section, we provide additional qualitative results under static scene settings. Static scenes typically encompass both indoor and outdoor environments featuring complex textures and geometric structures. These scenarios place strong emphasis on a model's ability to comprehend spatial layout and accurately model camera motion through the sequence.

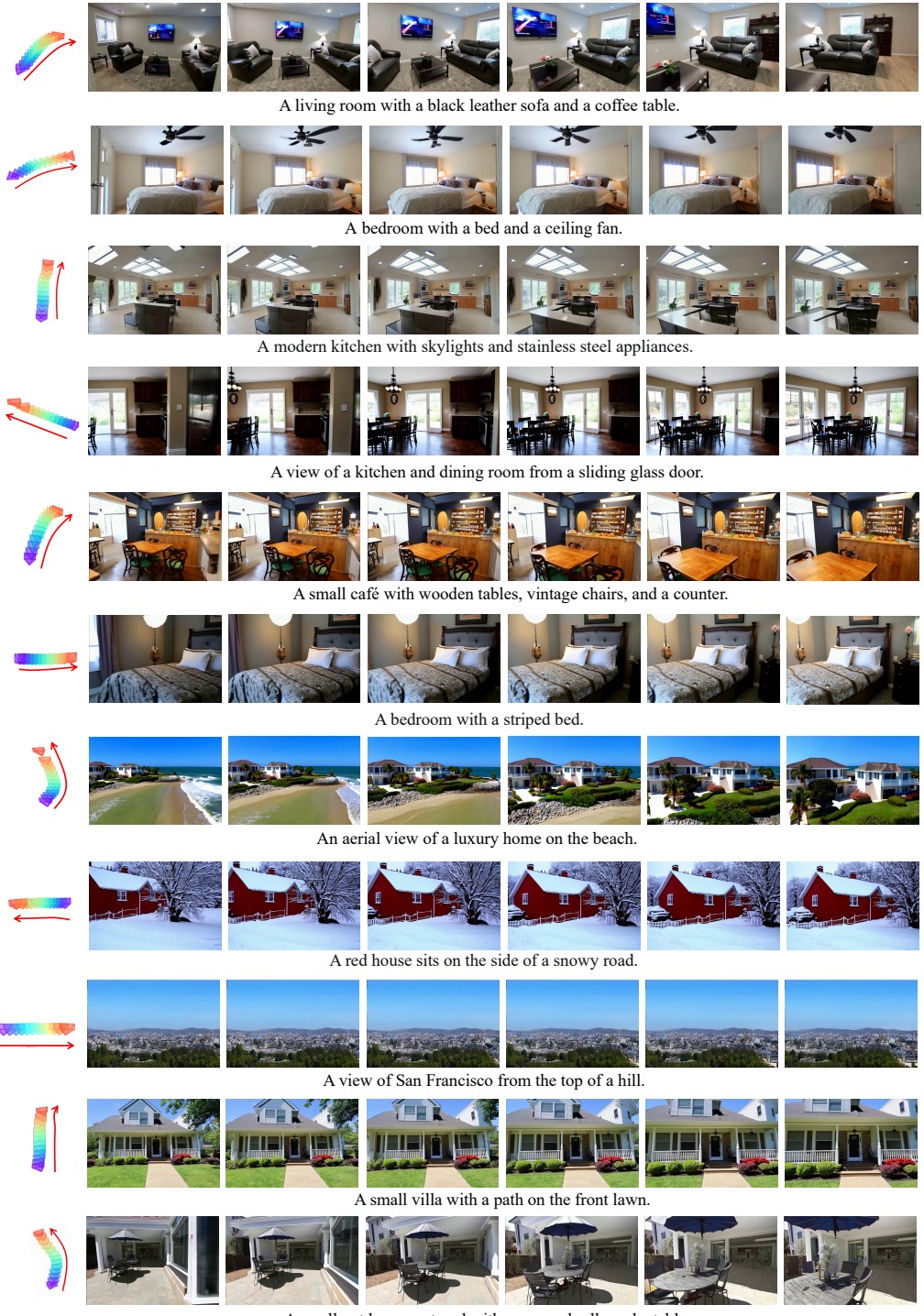

A living room with a black leather sofa and a coffee table.

A bedroom with a bed and a ceiling fan.

A modern kitchen with skylights and stainless steel appliances.

A view of a kitchen and dining room from a sliding glass door.

A small café with wooden tables, vintage chairs, and a counter.

A bedroom with a striped bed.

An aerial view of a luxury home on the beach.

A red house sits on the side of a snowy road.

A view of San Francisco from the top of a hill.

A small villa with a path on the front lawn.

A small outdoor courtyard with a sun umbrella and a table.

Figure 12: More qualitative results of MoCa on static scenes

## G  LIMITATIONS

The proposed MoCa, while demonstrating capability in generating high-quality camera-controlled videos, exhibits two certain limitations. Specifically, the current method primarily focuses on integrating camera control into text-to-video generation frameworks, without extending this capability to other multimodal data inputs. Future research should prioritize the idea of multimodal video generation capable of processing diverse input modalities, such as secondary editing of object regions and video style transfer applications. Secondly, although our approach successfully maintains object stability with camera movements, the current framework cannot precisely control where moving objects are positioned in the frame. For example, an object might unintentionally appear near the edges, resulting in less visual effect. Future work should address these challenges through enhanced object control generation and multimodal fusion techniques, ultimately aiming to achieve superior video generation quality with expanded creative possibilities.

