# OpenReview forum: "MoCa: Modeling Object Consistency for 3D Camera Control in Video Generation"
_ICLR.cc/2026/Conference — ICLR 2026 Poster_

### Official Review · Reviewer_uDhp · 2025-10-30

**Soundness:** 3
**Presentation:** 2
**Contribution:** 2
**Rating:** 4
**Confidence:** 4

**Summary:**

The paper proposes MoCa, a dual-branch diffusion-transformer framework for camera-controllable text-to-video generation. The framework targets to maintain consistency in terms of view, appearance and motion. Concretely: (1) a Spatial-Temporal Camera Encoder (ST-Encoder) encodes per-pixel camera rays using Plücker coordinates and fuses them via cross-attention into the DiT to improve view consistency; (2) a semantic guidance path (ReferenceNet) injects frozen vision-language features to stabilize appearance; (3) an object-aware motion disentanglement uses a high-frequency (2D-DWT) mask over VL features to separate local object motion from global camera motion for motion consistency. The model is fine-tuned from CogVideoX on RealEstate10K and evaluated on RealEstate10K (mostly static) and VidGen (dynamic): MoCa shows better or competitive camera controllability and object/background consistency (e.g., RotErr/OC/BC), and ablations support each component.

**Strengths:**

+Clear factorization of “consistency": Framing camera control around view/appearance/motion consistency is intuitive and aligns well with qualitative failures of prior work

+Appearance stabilization via VL features. The ReferenceNet “semantic guidance” improves object identity/texture stability;

+The high-frequency, object-aware mask improves OC/BC and motion plausibility when the camera is moving;

+Details are provided for reproducing the method: The paper lists training details and links to code/resources.

**Weaknesses:**

-Novelty vs. prior camera-conditioning is incremental.

The paper adopts Plücker coordinates for camera rays (as in CameraCtrl) and fuses conditions into DiT blocks (as in AC3D/VD3D). What is new in the ST-Encoder beyond adding spatial/temporal convs before cross-attention, and how does it differ from CameraCtrl’s/AC3D’s conditioning path are not addressed.

-Missing evaluation for proposed component for view consistency

It’s plausible that per-pixel rays + temporal convs should help view (and maybe motion) consistency, but the paper doesn’t directly measure “consistency” improvements from the ST-Encoder alone (e.g., removing temporal convs, or comparing addition vs. attention per consistency metric).

-Object-aware mask lacks important details/evaluation.

The mask comes from a 2D-DWT over foundation features, but key questions remain:
* How about multiple moving objects? How are several instance regions separated if VL features are class-level and not instance-level?

* Static vs moving distinction. How does the method decide that an object is static vs moving. Are static objects still routed through the disentangling branch; if so, any degradation?

* User-specified object motion. Can users specify independent object motion (e.g., “a bear walks left-to-right while the camera dollies forward”), or is motion always unsupervised/implicit? Motion-control scope and ability of the proposed disentanglement method are needed, given that it is one of the major contribution.

-Complexity/overhead not fully discussed.
The paper should report latency overhead from dual-branch fusion, and the cost/benefit of 2D-DWT during inference. Training uses 16×H200; inference speed vs. baselines would be helpful for practical adoption.

**Questions:**

Please see the Weakness section above.

---

> ### Author Response · Authors · 2025-11-24
>
> We greatly appreciate for your valuable suggestions and constructive comments. In response to the concerns raised, we have clarified the novelty of our camera condition module and provided ablations demonstrating clear gains in object  consistency. We also detailed how our motion disentanglement  handles multi-object and static/dynamic cases, and showed that our disentanglement enables user-specified object motion. Finally, we quantified the runtime overhead of each component.
> For clarity, all changes in the revised manuscript are highlighted in blue.
> **All of our visualizations and generated examples are provided on our project page at: https://anonymous.4open.science/w/MoCa-31E5/ .**
>
>
> ---
> > **W1:** Novelty vs. prior camera-conditioning is incremental: The paper adopts Plücker coordinates for camera rays (as in CameraCtrl) and fuses conditions into DiT blocks (as in AC3D/VD3D). What is new in the ST-Encoder beyond adding spatial/temporal convs before cross-attention, and how does it differ from CameraCtrl's/AC3D's conditioning path are not addressed.
>
> **Response:**  Thank you for raising this important question. Existing works mainly focus on improving camera accuracy. In contrast, we aim to maintain object consistency (view, appearance, and motion) under explicit 3D camera control. Specifically, our camera-conditioning mechanism differs from existing designs in two aspects:
>
> - Spatial-Temporal Convolution on Plücker Embedding: CameraCtrl (U-Net) models temporal relations of camera poses via temporal attention, while AC3D directly injects Plücker embeddings into DiT blocks via element-wise addition. Unlike prior works, we apply spatial-temporal convolutions to the Plücker embedding. This design enables the model to capture the camera trajectory as a continuous motion signal rather than discrete poses at the pixel-level. To verify this design, we conduct the ablation study on removing the spatial-convolution layers, temporal-convolution layers and both of them.
> As shown in Rows 1-3 vs. Row 5, removing spatial or temporal convolutions significantly degrades FVD, BC, and trajectory accuracy, while removing both yields the worst results. This confirms that explicit spatial-temporal modeling is essential for precise control and consistency, distinguishing our approach from other conditioning methods.
>
> - Attention-Based Fusion Mechanism: Unlike existing methods (e.g., CameraCtrl, AC3D) that use element-wise addition to inject camera conditions which often disrupts visual features. We employ an attention-based fusion strategy. This 'soft' modulation allows visual features to dynamically query camera conditions, preserving semantic integrity while ensuring pixel-level alignment. As shown in Row 4 vs. Row 5, our cross-attention mechanism outperforms addition fusion, achieving lower trajectory errors (TransErr, RotErr) and higher visual quality (CLIPSIM, BC, MS).
>
> | Dataset        | Method                      | FID ↓  | FVD ↓    | CLIPSIM ↑ | TransErr ↓ | RotErr ↓ | OC ↑  | BC ↑  | MS ↑ |
> |----------------|------------------------------|--------|----------|-----------|-------------|-----------|-------|-------|------------------------|
> | **RealEstate10K** | w/o Temporal Convolution      | 228.9  | 691.3    | 0.311     | 0.712       | 0.211     | 94.8%  | 95.2%  | 98.0%          |
> |                    | w/o Spatial Convolution      |  233.7 | 699.5  | 0.311     | 0.715       | 0.213     | 94.7%  | 95.3% |     98.2%      |
> |                    | w/o Spatial+Temporal Convs      |  239.2 |  729.6 |  0.309    |    0.717   |  0.215   | 94.3%  | 95.0%  |   97.9%        |
> |                | Element-Wise Addition        | 236.2  | 771.8    | 0.310     | 0.738       | 0.211     | 94.6%  | 95.1%  | 98.2%                   |
> |                | Ours (Attention-Based,Full)       | **207.4**  | **667.9**    | **0.312**     | **0.703**       | **0.208**     | **94.9%**  | **96.4%**  | **98.5%**             |
> | **VidGen**     | w/o Temporal Convolution      | 241.3  | 1672.8   | 0.347     | 0.730       | 0.088     | 94.2%  | 94.6%  | 97.7%                 |
> |                    | w/o Spatial Convolution      |  252.8 | 1702.9  |    0.347  |   0.728     |   0.085   |  94.4% | 94.3% |   98.1%        |
> |                    | w/o Spatial+Temporal Convs      | 275.8  | 1721.3  |  0.345    |   0.733    |  0.087    | 93.9%  | 94.1% |    97.4%       |
> |                | Element-Wise Addition        | 248.1  | 1738.6   | 0.345     | 0.732       | 0.084     | **94.9%**  | 94.7%  | 97.4%                   |
> |                | Ours (Attention-Based,Full)       | **232.2**  | **1643.7**   | **0.349**     | **0.724**       | **0.081**     | 94.7%  | **95.1%**  | **98.3%**              |

---

> > ### Author Response · Authors · 2025-11-24
> >
> > ---
> > > **W2:** Missing evaluation for proposed component for view consistency: It's plausible that per-pixel rays + temporal convs should help view (and maybe motion) consistency, but the paper doesn’t directly measure “consistency” improvements from the ST-Encoder alone (e.g., removing temporal convs, or comparing addition vs. attention per consistency metric).
> >
> > **Response:** We thank the reviewer for the constructive suggestion. We agree that verifying the contributions of the ST-Encoder components is essential. To address this, we conducted the suggested ablation studies on both RealEstate10K and VidGen.
> >
> > To enhance the view consistency, we adopt Plücker embedding to provide a geometrically interpretable representation that encodes camera trajectories at the pixel level. Meanwhile, our ST-Encoder employs spatial-temporal convolutions to model the spatial-temporal correlations of the camera sequence.
> >
> >
> > Furthermore, We compared our full model against two variants: (1) w/o Temporal Convs, and (2) Addition (replacing Attention with element-wise addition). The results are reported in the table in W1.
> >
> > - Effectiveness of Temporal Convolutions: As shown in Table W1 (Row 1 vs. Row 5), removing temporal convolutions ("w/o Temporal Convolution") leads to a significant degradation in FVD (e.g., +23.4 on RealEstate10K) and a drop in Motion Smoothness. This empirically confirms that the temporal convolutions in our ST-Encoder are critical for smoothing features across frames and suppressing flickering artifacts, directly validating the reviewer's intuition regarding motion consistency.
> >
> > - Effectiveness of Attention: Existing methods often use simplistic element-wise addition, which lacks pixel-wise interaction between camera conditions and visual features. Our experiments show that the Addition baseline results in inferior performance across most metrics. This demonstrates that the cross-attention mechanism effectively aligns per-pixel ray embeddings with visual features, ensuring that the generated views are both geometrically accurate and visually consistent.

---

> > > ### Author Response · Authors · 2025-11-24
> > >
> > > ---
> > > > **W3:** *Object-aware mask lacks important details/evaluation. The mask comes from a 2D-DWT over foundation features, but key questions remain:*
> > >
> > > ---
> > > > **W3.1:** *How about multiple moving objects? How are several instance regions separated if VL features are class-level and not instance-level?*
> > >
> > > **Response:** We thank the reviewer for this crucial question. We clarify that our high-frequency object-aware masking is capable of producing implicit structural masks for multiple objects, even when relying on class-level features.
> > >
> > > Although Vision-Language (VL) features are semantically class-level, multiple instances are typically spatially separated within the image plane. Our 2D-DWT extracts high-frequency components (such as edges and textures) from these features. These high-frequency signals naturally highlight the structural boundaries of distinct objects based on their spatial localizations, rather than relying solely on semantic instance labels. Consequently, the generated mask effectively separates multiple regions of interest. We recognize this mask as fine-grained structural information that are critical for precise localization of multi-object regions, which in turn helps disentangle local object motion from global camera movements.
> > >
> > > To validate this, we have added visualizations of object-aware masks for multi objects in Fig. 7 of our revised manuscript. Furthermore, we have conducted additional experiments on scenes with multiple moving objects. **The corresponding video results are available on our project page.** These results demonstrate our method's robustness in handling complex multi-object dynamics under camera control. For example, in the sample "The polar bear walking and a seal lying on the ice", our model generates two different objects and aligned with the text prompt. This showcases the effectiveness of our motion disentanglement in multi-object, multi-class scenarios.
> > >
> > > ---
> > > > **W3.2:** *Static vs. moving distinction. How does the method decide that an object is static vs. moving. Are static objects still routed through the disentangling branch; if so, any degradation?*
> > >
> > > **Response:**
> > > We thank the reviewer for the constructive comments. In our setting, dynamic scenes refer to cases where the foreground object has clear self-driven motion that is independent of the camera trajectory, while static scenes refer to videos where the foreground object has no own motion and only moves in the image plane because the camera is moving.
> > >
> > > **(1) No Explicit Distinction.** In our framework, we do not explicitly classify objects as static or dynamic. All objects (static or moving) are routed through the disentangling branch. Meanwhile, static objects do not exhibit any performance degradation.
> > >
> > > **(2) Why No Degradation Occurs?**  We clarify that the disentangling branch relies on high-frequency components (via 2D-DWT), which capture object structure, edges, and texture.
> > >
> > > - For dynamic objects. The combination of camera motion and object motion generates strong high-frequency signals, allowing the branch to effectively disentangle foreground motion from the background.
> > >
> > > - For static Objects. Even without self-driven motion, camera motion causes small shifts and scale changes in the image, which introduce high-frequency changes along the object edges. For the static scene, the high-frequency decomposition can still capture the structure and localization of static objects. Processing static objects through this branch ensures that their detailed appearance and boundaries are preserved against the global camera motion.
> > >
> > > We conducted both qualitative and quantitative studies to verify this. Specifically, for the static dataset RealEstate10K, we remove the ReferenceNet branch and train/infer our model using only a single-branch architecture. The results are as follows:
> > >
> > > - Qualitative Results: Without the ReferenceNet Branch, static objects exhibit noticeable appearance distortions and semantic misalignment with the text description. **The corresponding  results are shown in Figure 6 of our paper.**
> > >
> > > - Qualitative Results: As reported in the table below, removing the ReferenceNet Branch leads to noticeable drops in CLIPSIM, OC,  BC, and MS metrics in static scenes. This confirms that the dual-branch architecture helps maintain appearance consistency and text-vision alignment for static scenes.
> > >
> > >
> > > | Dataset          | Method                          | FID ↓ | FVD ↓   | CLIP-SIM ↑ | TransErr ↓ | RotErr ↓ | OC ↑   | BC ↑   | MS ↑   |
> > > |------------------|----------------------------------|--------|----------|-------------|-------------|-----------|---------|---------|---------|
> > > | **RealEstate10K**|  w/o ReferenceNet Branch       | 243.1  | 705.6    | 0.308       | 0.722       | **0.198**     | 94.1%   | 95.8%   | 97.9%   |
> > > |                  | Ours (Full, Attention Fusion)    | **207.4**  | **667.9**    | **0.312**       | **0.703**       | 0.208     | **94.9%**   | **96.4%**   | **98.5%**   |

---

> ### Author Response · Authors · 2025-11-24
>
> ---
> > **W3.3:** *User-specified object motion. Can users specify independent object motion (e.g., “a bear walks left-to-right while the camera dollies forward”), or is motion always unsupervised/implicit? Motion-control scope and ability of the proposed disentanglement method are needed, given that it is one of the major contribution.*
>
> **Response:** We appreciate the reviewer for highlighting this crucial aspect. Demonstrating independent control is indeed the best way to validate our disentanglement effectiveness.
>
> **(1) Can users specify independent object motion?** Yes! Users can explicitly specify object motion (e.g., direction, action) through text prompts, while independently controlling the camera via trajectory inputs. The scope of our motion control is defined as follows:
>
> - Object Motion: Driven by the semantic text prompt (e.g., "a bear walks left-to-right").
>
> - Camera Motion: Driven by the input camera pose sequence (e.g., "dolly forward").
>
> **(2) Handling Conflicting Motions (Proof of Disentanglement):**
> Our high-frequency disentanglement strategy ensures that the "object dynamics" (guided by text) are not overridden by or entangled with the "camera dynamics" (guided by pose).
>
> This is most evident in conflicting motion scenarios. For example, when the user specifies "a bird flying from right to left" while the camera pans to the right, our model correctly generated the bird moving against the camera flow. Without effective disentanglement, the strong global optical flow from the camera would typically force the object to drift with the background.  **We have added these specific "conflicting motion" demonstrations to our project page** to visually prove the motion-control scope and the robustness of our disentanglement method.
>
> To further validate this, we compare our model with AC3D under the same prompt and camera trajectory. We observe that they fail to realize the text-specified object motion direction because it cannot decouple object dynamics from camera dynamics. This contrast further confirms the necessity and effectiveness of our motion disentanglement strategy.
>
> We believe that this comparison clearly demonstrates the importance of our motion disentanglement design. We have included the corresponding video results on our project page and **highlighted this analysis in a separate section of the revised manuscript. (Sec. 4.4)**
>
> ---
> > **W4:** *Complexity/overhead not fully discussed. The paper should report latency overhead from dual-branch fusion, and the cost/benefit of 2D-DWT during inference. Training uses 16×H200; inference speed vs. baselines would be helpful for practical adoption.*
>
> **Response:** We thank the reviewer for emphasizing the importance of reporting model complexity and runtime overhead. We agree that this is an essential aspect, and **we have included this analysis in Sec. 4.7 of the revised manuscript.**
>
> (1) Inference Latency and Component Breakdown:
> We measured the inference latency on a single H200 GPU with BF16 precision over 50 denoising steps. The breakdown is provided in the table below.
> - Dual-Branch Fusion: This component adds 74.6s to the inference time. While this constitutes a notable portion of the total time, it is critical for maintaining appearance consistency and semantic alignment, which are the main bottlenecks in current camera-controllable generation.
>
> (2) 2D-DWT Module (Cost/Benefit): The 2D-DWT and hybrid condition fusion introduce a negligible overhead of only 9.73s (approx. 3.3% of total time). Despite its low cost, this module is the core of our disentanglement strategy. Removing it leads to significant drops in Object Consistency (OC) and Motion Smoothness (MS), confirming its high efficiency and necessity.
>
> (3) Training cost: MoCa is trained on 16 × H200 GPUs, with batch size = 64, for 20,000 iterations. This resource requirement is consistent with recent state-of-the-art video generation frameworks, ensuring the model learns robust motion priors from large-scale data.
>
> (4) Comparison with Baselines: We measure the inference time of AC3D, CogVideoX, and MoCa under the same setting. AC3D shares the same backbone CogVideoX as MoCa and thus provides a fair comparison. Although MoCa has a higher inference latency, it produce higher-quality, camera-controllable videos. For video generation tasks, users primarily care about the quality and controllability of the generated video. Current diffusion-based T2V methods generate a clip of only a few seconds takes several minutes, and this large mismatch between inference time and video length suggests that visual performance is far more important than inference speed.
> | Configuration  | Latency (s) | Overhead Contribution |
> |-----------------------------|-------------|------------------------|
> | Base Model (CogVideoX) | 104.9 | —  |
> | AC3D  | 224.6 | — |
> | + 2D-DWT & Hybrid Fusion | +9.73 | Low (~3.3%)   |
> | + Dual-Branch Fusion | +74.6 | Moderate (~25.6%)  |
> | Ours (Full Model) | 291.2 | Total |

---

### Official Review · Reviewer_fpxS · 2025-10-31

**Soundness:** 3
**Presentation:** 3
**Contribution:** 2
**Rating:** 6
**Confidence:** 3

**Summary:**

The paper tackles camera-controllable text-to-video (T2V) generation. The core claim is that smooth camera motion in 3D should manifest as consistent object view, appearance, and motion in 2D frames. MoCa proposes a dual-branch framework with:
(1) a Spatial-Temporal Camera Encoder using Plücker ray embeddings to inject geometry-aware camera signals for view consistency
(2) a semantic guidance path that feeds vision-language features from ReferenceNet into the denoiser to stabilize appearance
(3) an object-aware motion disentanglementto separate object dynamics from global camera motion for motion consistency

Experiments on RealEstate10K (static) and VidGen (dynamic) show improved camera control and object stability versus MotionCtrl, CameraCtrl, and AC3D, with ablations for each design choice.

**Strengths:**

- Paper is clearly written and easy to understand
- Motion disentanglement through DWT is neat. Additional discussions in the appendix help strengthen the authors' claims.
- Under a uniform 16-frame protocol, the method achieves top-rank or second-best mixes on RealEstate10K, and on VidGen improves key control/consistency metrics (RotErr, OC, CLIPSIM).
- Ablations explicitly test fusion choice (cross-attention vs addition) and discuss alternatives in the appendix, helping attribute gains to the proposed architecture rather than incidental training details. This strengthens causal claims about the design.

**Weaknesses:**

- Coverage of recent SOTA baselines is limited. While AC3D is included, other very recent transformer-based or geometry-aware camera-control methods cited in the text (e.g., VD3D, CamCo, ViewCrafter, CameraCtrl II) are not in the main comparison tables; this weakens claims of state-of-the-art across the latest literature.
- Camera accuracy evaluation relies on Mega-SAM reconstructions. It would help to quantify Mega-SAM failure rates or uncertainty propagation.
- The 16-frame evaluation setting seems to be a bit short in 2025. Recent models have shown to handle longer clips well; reporting an additional 32 to 48 frame setting (even on a subset) would better reflect practical camera control usage.

**Questions:**

In addition to the weakness section above:

- How sensitive is MoCa to camera path magnitude and frame count?

- What fraction of scenes yield Mega-SAM tracking failures, and how do you handle them?

- What is the runtime speed compared to the baselines?

---

> ### Author Response · Authors · 2025-11-24
>
> Thank you for your thorough reviews, insightful comments, and acknowledgment of our work.
> We expanded comparisons with recent methods, clarified camera evaluation reliability, and provided additional long-frame and complex-camera robustness evaluations. We also reported inference speed compared to the baseline, further strengthening the paper's claims.
> In the following, we provide detailed responses to the raised concerns and update the manuscript accordingly. All modifications have been highlighted in blue in the revised manuscript.
> **Our project page (https://anonymous.4open.science/w/MoCa-31E5/) shows all of our visualizations and generated examples.**
>
> ---
> > **W1:** Coverage of recent SOTA baselines is limited. While AC3D is included, other very recent transformer-based or geometry-aware camera-control methods cited in the text (e.g., VD3D, CamCo, ViewCrafter, CameraCtrl II) are not in the main comparison tables; this weakens claims of state-of-the-art across the latest literature.
>
> **Response:** We thank the reviewer for suggesting these recent baselines. We agree that VD3D, CamCo, ViewCrafter, and CameraCtrlII are significant recent contributions to the field. However, we primarily focused on MotionCtrl, CameraCtrl, and AC3D in our main paper, as they are currently the most widely used open-source benchmarks that allow for fair and reproducible quantitative evaluation.
>
> Regarding the additional methods mentioned: CamCo and CameraCtrlII have not released their code or pre-trained models, making reproduction infeasible. Nevertheless, to address the reviewer's concern, **we have added qualitative comparisons with VD3D and ViewCrafter to our project page:**
>
> - VD3D: Since the code and trajectory parameters are unavailable, we approximated the camera poses based on their official demo videos. As shown on our project page, our method achieves superior motion consistency and supports significantly longer generation (49 frames vs. VD3D's 16 frames).
>
> - ViewCrafter: As ViewCrafter is an Image-to-Video (I2V) method (unlike our T2V approach), we used the first frame generated by our model as input for ViewCrafter to enable a comparison. The results demonstrate that our method surpasses ViewCrafter in visual realism and motion coherence.
>
> ---
> > **W2 & Q2:** Camera accuracy evaluation relies on Mega-SAM reconstructions. It would help to quantify Mega-SAM failure rates or uncertainty propagation.
>
> **Response:** Thank you for raising this important point. It is difficult to define a clear  "failure rate"  for camera pose estimation. In the Mega-SAM paper, the reported metrics for camera pose accuracy (ATE, RTE, RRE) all measure the discrepancy between the predicted camera trajectory and the ground truth, but these are measures of "error" rather than "failure".  To decide whether a case is a "failure", one has to choose thresholds on these errors, which is highly subjective.  There is  no standard method for defining and measuring such failure cases.
>
> For our experiments, on each dataset (RealEstate10K and VidGen), we randomly sample 100 samples (the GT videos together with camera trajectories estimated by Mega-SaM). We visualize the estimated camera trajectories and scene construction. We manually judge the failure cases. We mark a case as a failure when the predicted camera trajectories are extremely noisy or clearly inconsistent with the camera motion observed in the GT video, and when the reconstructed  scene exhibits severe inconsistencies. Based on this protocol, the estimated failure rate is **0%** on the static scene dataset RealEstate10K and **2%** on the dynamic scene dataset VidGen. In addition, during the construction of our test set, we have already manually removed such failure cases.
>
> As the prior work[1] has noted, such failure cases mainly arise from two cases:
> - The foreground object occupies a large portion of the frame, leaving the model with insufficient stable feature points for reliable tracking.
> - The camera motion is approximately colinear with the object motion (e.g., selfie scenarios), making it difficult for the model to distinguish camera movement from foreground object movement.
>
> These two cases do not occur in static scene dataset RealEstate10K and are extremely rare in the dynamic dataset VidGen.
>
> [1] Zhoutong Zhang, Forrester Cole, Zhengqi Li, Michael Ru- binstein, Noah Snavely, and William T Freeman. Structure and motion from casual videos. In Proc. European Conf. on Computer Vision (ECCV), 2022. 1, 2, 5, 6, 7, 8.

---

> ### Author Response · Authors · 2025-11-24
>
> ---
> > **W3:** The 16-frame evaluation setting seems to be a bit short in 2025. Recent models have shown to handle longer clips well; reporting an additional 32 to 48 frame setting (even on a subset) would better reflect practical camera control usage.
>
> **Response:** Thank you for your detailed suggestions about the evaluation of long-frame generation. Indeed, our method is able to generate much longer sequences (49 frames) compared with CameraCtrl and MotionCtrl, both of which are limited to 16-frame outputs. For a fair comparison, we uniformly downsample our generated videos to evaluate quantitative results in our paper. All examples showcased on our project page are 49-frame generated videos.
>
> We further evaluate our model at 49-frame setting on both datasets. The results are shown in the table below. We can observe that increasing the video length does not  affect the visual quality of the generated videos, with all metrics remaining relatively stable.  RotErr and TransErr measure cumulative errors over the video frame sequence. Meanwhile, our generation results significantly outperform the baseline CogVideoX.
>
> | Dataset          | Method             | FID ↓ | FVD ↓ | CLIPSIM ↑ | TransErr ↓ | RotErr ↓ | OC ↑   | BC ↑   | MS ↑   |
> |------------------|--------------------|--------|--------|-------------|------------|-----------|---------|---------|---------|
> | **RealEstate10K** | baseline (49 frames) | 223.4 | 671.2 | 0.312       | -          | -         | 94.4%   | 95.5%   | 98.4%  |
> |                  | ours (16 frames)     | 207.4 | 667.9 | 0.312       | 0.703      | 0.208     | 94.9%   | 96.4%   | 98.5%   |
> |                  | ours (49 frames)     | 216.3 | 669.1 | 0.312       | 1.629      | 0.507     | 94.8%   | 95.7%   | 98.4%   |
> | **VidGen**                 | baseline (49 frames) | 221.4 | 1673.9 | 0.347      | -          | -         | 93.8%   | 94.1% | 97.5%   |
> |                  | ours (16 frames)     | 232.2 | 1643.7 | 0.349      | 0.724      | 0.081     | 94.7%   | 95.1%   | 98.3%   |
> |                  | ours (49 frames)     | 237.6 | 1659.8 | 0.350      | 1.791      | 0.169     | 94.9%   | 94.8%   | 98.1%   |
>
>
> ---
> > **Q1:** How sensitive is MoCa to camera path magnitude and frame count?
>
> **Response:** We thank the reviewer for this insightful question. We are happy to report that MoCa is robust to variations in both camera path magnitude and frame count.
>
> - Regarding Camera Path Magnitude: Our model generalizes well to diverse trajectory patterns. To validate this, **we have provided visualizations of several out-of-distribution and complex camera trajectories on our project page.** These results demonstrate that MoCa maintains high visual quality and precise controllability even under large or irregular camera movements.
>
> - Regarding Frame Count: Our method maintains consistent performance across different sequence lengths. As shown in our quantitative evaluations (refer to W3) and the 49-frame demos on our project page, MoCa preserves excellent generation quality and temporal coherence as the number of frames increases, effectively handling longer video generation without degradation.
>
>
> ---
> > **Q3:** The runtime speed compared to the baselines.
>
> **Response:** We measure the inference latency on a single H200 GPU with BF16 precision over 50 denoising steps. The results are as follows:
>
> | Configuration            | Latency (s) |
> |--------------------------|-------------|
> | Base Model (CogVideoX)   | 104.9       |
> | AC3D       | 224.6       |
> | Ours (Full Model)        | 291.2       |
>
> We report AC3D's latency since it uses the same backbone CogVideoX as MoCa, allowing a fair comparison of runtime overhead. While MoCa introduces additional inference time due to the dual-branch architecture, we prioritize generation quality and controllability over inference speed. As demonstrated in the quantitative comparisons with AC3D and with the backbone CogVideoX, our method achieves significant improvements in critical metrics (OC, BC, MS, FVD). We believe this trade-off is well-justified, as the primary goal in video generation is to produce high-quality, precisely controllable content rather than maximizing inference speed.

---

### Official Review · Reviewer_QNWA · 2025-11-01

**Soundness:** 3
**Presentation:** 3
**Contribution:** 3
**Rating:** 6
**Confidence:** 3

**Summary:**

This paper introduces MoCa, a framework for camera-controlled video generation that improves object stability by focusing on maintaining consistency in view, appearance, and motion. The model uses a dual-branch architecture with semantic guidance to preserve object identity and a disentanglement mechanism to separate object dynamics from camera movement.

**Strengths:**

1. The video examples shown in the paper supp are strong. The generated videos look much more stable and realistic than the comparison methods.
2.  The main idea of framing the problem around "object consistency" (view, appearance, and motion) is a smart way to tackle the challenge. It breaks down a complex 3D problem into more manageable 2D properties that we can observe in the final video.
3.  The method for separating object motion from camera motion is clever. Using a 2D Discrete Wavelet Transform (2D-DWT) to create a "high-frequency object-aware mask" is an interesting technical contribution.

**Weaknesses:**

1. The paper relies on Object Consistency (OC) and Background Consistency (BC) scores from VBench to prove its main contribution. However, as the VBench paper itself explains, these metrics just measure feature similarity (using DINO and CLIP) across frames. This means they mainly check if an object is consistently present, not if its motion is natural or if it's free from distortion. A video with a "frozen" object sliding unnaturally across the screen could still get a high OC score, which doesn't really support the claim of improved motion consistency.
2. The method is built by fine-tuning CogVideoX, a very large (~5B) foundation model. While this helps achieve impressive results, it makes it hard to judge how much of the performance comes from the new MoCa architecture versus the power of the base model, especially considering the nature of randomness in the generation.
3. The paper could use another round of proofreading. There are several places where the citation formatting is incorrect (e.g., in Section 4.1, should be a proper \citep command). These small errors, along with some sections that are a bit dense, can make the paper harder to read and feel less polished, such the following part:

   - The semantic guidance strategy (Section 3.2) uses a ReferenceNet to maintain object identity. The paper says it uses "reference video frames" (line 228), but it's not clear what this means in practice. Is the input just the first frame of the video, or a specific set of keyframes? This detail is crucial for understanding how the model gets its "identity guidance."
   - The high-frequency object-aware mask is a key part of the motion disentanglement. Could you provide more detail on how this mask is actually used in the "Hybrid Condition Fusion" step (line 263)? For example, is it used as a soft attention map to guide the DenoisingNet, or is it combined with other features in a different way? A clearer explanation of this mechanism would be very helpful.

**Questions:**

- The semantic guidance strategy (Section 3.2) uses a ReferenceNet to maintain object identity. The paper says it uses "reference video frames" (line 228), but it's not clear what this means in practice. Is the input just the first frame of the video, or a specific set of keyframes? This detail is crucial for understanding how the model gets its "identity guidance."
   - The high-frequency object-aware mask is a key part of the motion disentanglement. Could you provide more detail on how this mask is actually used in the "Hybrid Condition Fusion" step (line 263)? For example, is it used as a soft attention map to guide the DenoisingNet, or is it combined with other features in a different way? A clearer explanation of this mechanism would be very helpful.

---

> ### Author Response · Authors · 2025-11-24
>
> We extend our sincere thanks for your comprehensive review and positive assessment of our work.
> In this rebuttal, we addressed your concerns by clarifying the roles and limitations of VBench metrics (OC and BC), providing additional evaluations (Motion Smoothness), and demonstrating the contribution of our MoCa architecture beyond the base model. We also resolved questions regarding semantic guidance, the usage of the high-frequency mask, corrected all citation formatting, and revised the manuscript accordingly.
> All modifications have been highlighted in blue in our revised manuscript.
> **Our visualizations can be found on our project page: https://anonymous.4open.science/w/MoCa-31E5/ .**
>
> ---
> > **W1:** *The paper relies on Object Consistency (OC) and Background Consistency (BC) scores from VBench to prove its main contribution. However, as the VBench paper itself explains, these metrics just measure feature similarity (using DINO and CLIP) across frames. This means they mainly check if an object is consistently present, not if its motion is natural or if it's free from distortion. A video with a "frozen" object sliding unnaturally across the screen could still get a high OC score, which doesn't really support the claim of improved motion consistency.*
>
>
> **Response:**
> We thank the reviewer for this insightful comment. We agree that your viewpoint that OC (Object Consistency) and BC (Background Consistency) in VBench primarily measure feature similarity rather than the naturalness of motion. While OC and BC are not intended to assess  unnatural motion (e.g., a "frozen" sliding object) directly, they remain crucial for evaluating view and appearance stability. As demonstrated in Fig. 8A and Fig. 9A of the VBench paper [1], these metrics are highly sensitive to presence and appearance inconsistency in both the foreground object (OC) and the background scene (BC). Therefore,  our high scores on OC and BC demonstrate that our architecture effectively maintains the object view consistency and apearance consistency.
>
> To directly address the concern regarding "frozen" objects and evaluate motion consistency and naturalness, we have conducted additional evaluation using the Motion Smoothness (MS) metric from VBench on Realestate10K and VidGen. This metric measures whether the motion in the generated video is smooth and natural. We consider it a complementary measure of motion consistency. Meanwhile, the CLIPSIM metric shows reduced performance for "frozen" objects whose motion is inconsistent with the textual description.
>
> As shown in the table below, our method achieves superior CLIPSIM and Motion Smoothness (MS) scores compared to other methods. These results mean that our method maintains stable motion in dynamic scenes, whereas other approaches show notable degradation. We have clarified the distinct roles of OC, BC and MS, and updated the manuscript to explain these metrics.
>
> | Dataset       | Method           | CLIPSIM ↑ | OC ↑  | BC ↑  | MS ↑ |
> |--------------|------------------|------------|-------|-------|-------|
> | **RealEstate10K** | MotionCtrl       | 0.309      | 94.6%  | 95.8%  | 97.8% |
> |              | CameraCtrl       | 0.305      | 94.3%  | 94.7%  | 97.7% |
> |              | AC3D             | 0.309  | **95.1%** | 95.3%  | 98.5% |
> |              | Ours             | **0.312**  | 94.9%  | **96.4%** | **98.5%** |
> | **VidGen**       | MotionCtrl       | 0.333      | 92.6%  | 93.2%  | 97.1% |
> |              | CameraCtrl       | 0.339      | 92.9%  | 93.1%  | 96.9% |
> |              | AC3D             | 0.345      | 93.5%  | 94.7%  | 97.7% |
> |              | Ours             | **0.349**  | **94.7%** | **95.1%** | **98.3%** |
>
> [1] Ziqi Huang, Yinan He, Jiashuo Yu, Fan Zhang, Chenyang Si, Yuming Jiang, Yuanhan Zhang, Tianxing Wu, Qingyang Jin, Nattapol Chanpaisit, et al. VBench: Comprehensive Benchmark Suite for Video Generative Models. In Proceedings of the IEEE/CVF Conference on Computer Vision and Pattern Recognition, pp. 21807–21818, 2024.

---

> ### Author Response · Authors · 2025-11-24
>
> ---
> > **W2:**  *The method is built by fine-tuning CogVideoX, a very large (~5B) foundation model. While this helps achieve impressive results, it makes it hard to judge how much of the performance comes from the new MoCa architecture versus the power of the base model, especially considering the nature of randomness in the generation.*
>
> **Response:**
> We appreciate the reviewer's concern regarding the attribution of performance gains.  To evaluate the contribution of MoCa, we utilized the baseline CogVideoX (T2V) to generate 49-frame videos directly from text prompts. For our method, we used the same text prompts and fixed random seeds combined with random camera control sequences (since the baseline lacks camera controllability). This setup ensures that performance difference is directly attributable to our proposed architecture rather than the base model's capabilities or nature of randomness in the generation.
>
> - Quantitative Analysis: As shown in the table below, since the baseline cannot accept camera poses, we exclude trajectory-related metrics (TransErr, RotErr) and focus on video quality and consistency. The results demonstrate that our method outperforms the baseline in Object Consistency (OC), Background Consistency (BC), and Motion Smoothness (MS), particularly in dynamic scenes (VidGen).
>
> - Qualitative Evidence: **We have added visual comparisons on our project page.** Our method produces natural motion with superior view and appearance consistency.
>
> | Dataset        | Method        | FID ↓  | FVD ↓   | CLIPSIM ↑ | OC ↑    | BC ↑    | MS ↑    |
> |----------------|---------------|--------|---------|-------------|---------|---------|---------|
> | **RealEstate10K** | Baseline(49-frame)       | 223.4 | 671.2   | 0.312       | 94.4%  | 95.5%  | 98.4%  |
> |                | Ours (49-frame) | **216.3** | **669.1**   | **0.312**       | **94.8%**  | **95.7%**  | **98.4%**  |
> | **VidGen**        | Baseline(49-frame)         | **221.4** | 1673.9  | 0.347       | 93.8%  | 94.1%  | 97.5%  |
> |                | Ours (49-frame) | 237.6 | **1659.8**  | **0.350**       | **94.9%**  | **94.8%**  | **98.1%**  |

---

> ### Author Response · Authors · 2025-11-24
>
> ---
> > **W3:**  *The paper could use another round of proofreading. There are several places where the citation formatting is incorrect (e.g., in Section 4.1, should be a proper \citep command). These small errors, along with some sections that are a bit dense, can make the paper harder to read and feel less polished.*
>
> **Response:**  We thank the reviewer for the detailed reading and for pointing out the citation formatting errors  and the lack of clarity in certain sections.  In the revised version of the paper, we have carefully corrected all citation commands (line 286, 287, 289, 296, 299) and improved the clarity and readability of relevant sections . Additionally, we have re-clarified the usage of ReferenceNet in our revised manuscript (line 230). All modifications have been highlighted in blue.
>
> ---
> > **W3.1 & Q1.1:**  *The semantic guidance strategy (Section 3.2) uses a ReferenceNet to maintain object identity. The paper says it uses "reference video frames" (line 228), but it's not clear what this means in practice. Is the input just the first frame of the video, or a specific set of keyframes? This detail is crucial for understanding how the model gets its "identity guidance."*
>
> **Response:** We apologize for any confusion caused by the unclear description in this section. The phrase “the information of the reference video frames” was intended to refer to the implicit feature representations extracted by the ReferenceNet, rather than raw pixel-level frames. The implicit information serves as a stable guidance for the whole scene.  In practice, we extract the Vision-Language (VL) features from the visual branch in each DiT block of the ReferenceNet and inject them into the DenoisingNet. We consider that these VL features are aligned in both the visual and semantic spaces. We have clarified this in the revised manuscript (line 230).
>
> ---
> > **W3.2 & Q1.2:** *The high-frequency object-aware mask is a key part of the motion disentanglement. Could you provide more detail on how this mask is actually used in the "Hybrid Condition Fusion" step (line 263)? For example, is it used as a soft attention map to guide the DenoisingNet, or is it combined with other features in a different way? A clearer explanation of this mechanism would be very helpful.*
>
> **Response:** Thank you for pointing this out. The high-frequency mask effectively distinguishing foreground objects from the background. In the "Hybrid Condition Fusion" step, the mask is employed to dynamically modulate the attention mechanism:
>
> - Cross-Attention Interaction: The object-aware mask is integrated into the multi-head cross-attention layers. Specifically, it interacts with the visual features to guide the generation of attention maps. This allows the DenoisingNet to selectively attend to foreground regions (highlighted by the high-frequency signals) while still retaining the spatial context of the background.
>
> - Temporal Refinement: Following this spatial fusion, we apply self-attention along the temporal dimension. This ensures that the motion patterns disentangled by the mask are propagated coherently across frames, improving temporal consistency.

---

### Author Response · Authors · 2025-11-28
**Summary of reviewer comments and request for area chair consideration**

Dear area chair,

We sincerely appreciate the time and effort you have dedicated to our paper and believe that we have thoroughly addressed all of the reviewers' concerns. In this work, we propose MoCa, a dual-branch framework for camera-controllable text-to-video generation. MoCa learns view, appearance, and motion consistency to bridge 3D camera movement and its corresponding changes in 2D frames, achieving high quality camera-controllable video generation. Extensive quantitative and qualitative experiments show that MoCa outperforms other methods, particularly in dynamic scenes. All visualizations in the paper and the rebuttal are provided on our project page: https://anonymous.4open.science/w/MoCa-31E5/ . The implementation code is available at: https://anonymous.4open.science/r/MoCa-31E5/ .
## Response to Reviewer QNWA
In our response to reviewer QNYA, we have summarized the points we have addressed:
- **Evaluation** of motion naturalness and clarification of OC and BC metrics
- **Evaluation** of performance gains from MoCa rather than the backbone CogVideoX
- **Detailed explanation** of the ReferenceNet and the Hybrid Condition Fusion

In the rebuttal, we additionally introduced the Motion Smoothness (MS) Score from VBench for motion naturalness, clarified that OC/BC mainly focus on view and appearance consistency, and the CLIPSIM also reflects motion consistency through semantic alignment. Meanwhile, MoCa achieves the best MS and CLIPSIM among all compared methods, while also maintaining strong OC/BC scores **(W1)**. We further evaluated MoCa’s contribution by comparing it against the backbone CogVideoX, showing consistent improvements in both quantitative and qualitative experiments **(W2)**. To more clearly explain the ReferenceNet and the Hybrid Condition Fusion, we provided detailed descriptions and corrected all citation issues in the revised manuscript **(W3 & Q1)**.
## Response to Reviewer fpxS
In our response to reviewer fpxS, we have summarized the points we have addressed:
- **Coverage** of recent camera-controllable video generation methods
- **Evaluation** of long-clip generation ability
- **Robustness** to complex camera trajectories and varying frame counts
- **Reliability of Mega-SAM** reconstruction for camera trajectory evaluation
- **Evalution** of runtime speed compared to the baselines

In our rebuttal, we provided qualitative comparisons to VD3D and ViewCrafter on our project page **(W1)**. To address the long-clip concern, we reported 49-frame quantitative results on both datasets, showing that MoCa maintains stable performance from 16 to 49 frames. It is noteworthy that all visualizations are presented in the 49-frame setting **(W3 & Q1)**. We further provided visualizations under complex camera trajectories to demonstrate robustness to camera-path magnitude **(Q1)**. To evaluate the reliability of Mega-SAM reconstruction, we quantified Mega-SAM failure rates and summarized the typical reconstruction failure modes **(W2 & Q2)**. We also reported an analysis of inference latency in our rebuttal **(Q3)**.
## Response to Reviewer uDhp
In our response to reviewer uDhp, we have summarized the points we have addressed:
- **Detailed explanation** of our camera-conditioning mechanism and its strengths over other Plücker-based camera modules
- **Contribution of different components in ST-Encoder**
- **Extended evaluation of the object-aware mask,** highlighting its benefits for multi-object scenes, static scenes, and user-specified motion control
- **Analysis of model complexity and overhead**

In our rebuttal and revised manuscript, we clarified the novelty of our camera-conditioning mechanism: it applies spatial-temporal convolutions on Plücker embedding and an attention-based fusion instead of simple addition **(W1)**. To evaluate the contribution of our camera condition module, we added ablations that remove spatial convolutions, temporal convolutions, both, or replace attention with element-wise addition, showing clear gains in OC/BC/MS on RealEstate10K and VidGen **(W1 & W2)**. We further strengthened the evaluation of the 2D-DWT object-aware mask by adding experiments on multi-object scenes. To address the concern about static scenes, we conducted an ablation study, confirming that the disentangling branch brings clear improvements in static-scene generation. To validate user-specified and even conflicting motion cases (e.g., object and camera moving in opposite directions), we conducted a detailed qualitative analysis in **a new Section 4.4 of the revised manuscript**, highlighting the effectiveness of our object-aware mask. All corresponding visualizations have been updated on our project page **(W3)**. We also introduced **a separate Section 4.7** that presents a detailed analysis of complexity and benefit, training cost, and inference speed **(W4)**.

We hope that you will consider this context in your evaluation of our paper.

Thank you!

---

### Meta-Review · Area_Chair_2Qii · 2026-01-04

**Summary:**

This paper explores the camera control in text-to-video generation and presents MoCa, a dual-branch framework. MoCa bridges the gap between the 2D pixel space and the underlying 3D world by modeling object consistency to implicitly learn 3D relationships between camera and scene. In the initial review, this paper is rated as 6,6 and 4. The concerns mainly lie in the evaluation metrics, the comparison with previous works, additional details and ablation studies, and writting. During rebuttal, the authors provides explanation and additional qualitative and quantitative results, demonstrating the effectiveness of the proposed method. Therefore, AC confirms that the reviewers' concerns are mostly addressed and recommends this paper as **Accept (poster)**.

**Reviewer Concerns:**

The initial concerns mainly lie in the evaluation metrics, the comparison with previous works, additional details and ablation studies, and writting.

**Evaluation metrics**: authors evaluate the motion smoothness, and they also analyze the reliability of Mega-SAM. This main concern is mostly addressed.

**The comparison with previous works**: authors provides additional comparison with MotionCtrl, CameraCtrl and AC3D on both RealEstate10K and VidGen benchmarks, under 16 frames and 49 frames settings. This main concern is mostly addressed.

**Additional details and ablation studies**: authors provides ablation studies and analysis on spatial and temporal conv, ReferenceNet branch, and the backbones. They also analyze the computational cost of the proposed modules. This main concern is mostly addressed.

**Reviewer Scores:**

**Reviewer QNWA** keeps 6. Reviewer QNWA do not have chance to reply, but the concerns are mostly addressed.

**Reviewer fpxS** keeps 6. Reviewer fpxS do not have chance to reply, but the concerns are mostly addressed.

**Reviewer uDhp** raises to 6. Reviewer uDhp do not have chance to reply, but the concerns are mostly addressed.

---

### Decision · Program_Chairs · 2026-01-26

Accept (Poster)